# A method for using unmanned aerial vehicles for emergency investigation of single geo-hazards and sample applications of this method

**Haifeng Huang[1, 2, 3], Jingjing Long[2, 3], Wu Yi[3, 4], Qinglin Yi[3, 4], Guodong Zhang[3, 4], Bangjun Lei[1]**

[1] Hubei Key Laboratory of Intelligent Vision Based Monitoring for Hydroelectric Engineering, China Three Gorges University, Yichang 443002, China

[2] Key Laboratory of Disaster Prevention and Mitigation of Hubei Province, China Three Gorges University, Yichang 443002, China

[3] National Field Observation and Research Station of Landslides in the Three Gorges Reservoir Area of Yangtze River, China Three Gorges University, Yichang 443002, China

[4] Collaborative Innovation Center for Geo-Hazards and Eco-Environment in The Three Gorges Area of Hubei province, Yichang 443002, China

Correspondence to: Jingjing Long (arielljj@foxmail.com)

**Abstract.** In recent years, unmanned aerial vehicles (UAVs) have become widely used in emergency investigations of major natural hazards over large areas; however, UAVs are less commonly employed to investigate single geo-hazards. Based on a number of successful investigations in the Three Gorges Reservoir area, China, a complete UAV-based method for performing emergency investigations of single geo-hazards is described. First, a customized UAV system that consists of a multi-rotor UAV subsystem, an aerial photography subsystem, a ground control subsystem and a ground surveillance subsystem is described in detail. The implementation process, which includes four steps, i.e., indoor preparation, site investigation, on-site fast processing and application, and indoor comprehensive processing and application, is then elaborated, and two investigation schemes, automatic and manual, that are used in the site investigation step are put forward. Moreover, some key techniques and methods, e.g., the layout and measurement of ground control points (GCPs), route planning, flight control and image collection, and the Structure from Motion (SfM) photogrammetry processing, are explained. Finally, three applications are given. Experience has shown that using UAVs for emergency investigation of single geo-hazards greatly reduces the time, intensity and risks associated with on-site work and provides valuable, high-accuracy, high-resolution information that supports emergency responses.

**Keywords.** single geo-hazard; landslide; emergency investigation; unmanned aerial vehicle (UAV); emergency response

## 1 Introduction

The aim of the emergency investigation of geo-hazards is to provide basic and essential information, including the characteristics of disasters, damages and environmental conditions, for use in emergency decision-making and effective response. These investigations are therefore a top priority, and the speed and efficiency of the implementation process and

the accuracy of the results must be further improved (Liu, 2006; Liu et al., 2010; Lu and Xu, 2014). In general, traditional
methods of emergency investigation of geo-hazards are used; i.e., specialists inspect the affected area with cameras and simple measurement tools, then report their conclusions based on the field investigation and professional knowledge. There is no doubt that these efforts using the traditional method require substantial manpower, long working hours and highly intense work; moreover, they often face difficulties because certain parts of geo-hazards, such as high cliffs or areas covered by lush vegetation, are inaccessible to humans. In particular, on-site investigators must contend with the considerable risks
associated with additional disasters that may occur during the process of the emergency investigation. In addition, the conclusions of these investigations are often inaccurate because they are primarily local, qualitative and speculative. Even some quantitative results, such as the length, width, or area of a geo-hazard, may deviate strongly from the actual situation. Therefore, relying solely on traditional ground-based emergency investigation methods inevitably reduces the efficiency and effectiveness of decision-making and the responses associated with geo-hazard emergencies.

Remote sensing is fast and covers large areas at high resolution, and it has considerable advantages in the field of emergency investigation of major natural hazards (Joyce et al., 2009; Boccardo and Tonolo, 2015). Along with the rapid development of unmanned aerial vehicle (UAV) remote sensing technology, it has been widely used in mapping (Aicardi et al., 2015), environmental investigation (Aicardi et al., 2016) and emergency investigation (Boccardo et al., 2015). Remote sensing from UAVs has been especially widely used in the emergency investigation of geo-hazards given its unique
advantages, such as low cost, ease of operation, minimal risk and efficient image acquisition (Lewis, 2007; Adams et al., 2014; Li et al., 2014; Fernandez Galarreta et al., 2015). For example, in the USA, UAVs were used to perform damage inspections after Hurricane Katrina (Pratt et al., 2006) and Hurricanes Wilma and Ike (Steimle et al., 2009). In Taiwan, a helicopter UAV was used to collect imagery to support post-disaster reconnaissance, disaster restoration and reconstruction assessments after Typhoon Morakot (Chou et al., 2010). In addition, UAVs have gradually become an indispensable means
of disaster investigation and assessment after earthquakes, e.g., the Wenchuan earthquake in 2008 (Zhou et al., 2008), the L'Aquila earthquake in 2009 (Quaritsch et al., 2010), the Haiti earthquake in 2010 (Huber, 2010), the Japan earthquake in 2011 (Ackerman, 2011), and the Lushan earthquake in 2013 (Xu et al., 2014). However, the above applications show that UAVs are mainly used in emergency investigations or loss assessments associated with major natural hazards, e.g., earthquakes or their secondary geo-hazards (landslides and rock collapses) that cover large areas. These UAV systems are
usually large, complex and costly, and the acquisition of the final results is a very involved process that requires large amounts of time. Moreover, of the large number of geo-hazards that occur annually, "single" disasters with limited spatial extents (such disasters usually have an area that extends from a few hundred to several million square metres) and limited volumes (which usually extend from a few thousand to tens of millions of cubic metres) account for the vast majority. For example, in 2015, a total of 8,224 geo-hazards (of which most were landslides) occurred in the mainland area of China. Of
these events, 8,180 were medium- (0.1-1.0 million $m^3$) and small-(less than 0.1 million $m^3$) sized. These events accounted for 99.5% of the total number of events, and the direct economic losses from these events were 200 million USD, which represents 55.8% of the total direct losses (Mlr P.R. China, 2015). On the one hand, it can be seen that the emergency investigation of single geo-hazards and the response to such events is very necessary for the prevention and mitigation of disasters. On the other hand, because of the potentially greater losses and huge amounts, efficient and effective methods are

typically required; e.g., only a few days or even hours are usually available to propose the response measures. In this case, a simple, flexible and small UAV system, as well as a rapid, efficient on-site image acquisition method and UAV-based remote sensing results processing method must be used to ensure that the whole airborne-based emergency investigation procedures can be completed in the shortest possible time, thus providing valuable information for subsequent efforts, such as ground-based investigation or the design of emergency responses. However, no complete, systematic and effective method of

using UAVs for the investigation of single geo-hazards presently exists, and many challenges, such as the lack of customized UAV systems and the lack of sound on-site implementation methods, are hampering the use of UAVs in specific applications.

The main aim of carrying out this study is to describe a number of successful examples of using UAVs to perform emergency investigation of geo-hazards in the Three Gorges Reservoir area of China in recent years and to establish a complete method of using UAVs for emergency investigation of single geo-hazards. This method includes a customized UAV,

the implementation of UAV-based investigation, the key techniques and methods used during the investigation and the processing of the results. In addition, three applications are provided to demonstrate the applicability of the proposed method.

## 2 Customized UAV

Most single geo-hazards (which mainly include landslides, rock collapses and debris flows) that require emergency

investigation, although they are generally of medium or small size, are often located in mountainous areas with rugged topography, where only a limited area can be observed from the ground, and that experience changeable meteorological conditions, e.g., uncertain wind speed and direction. In addition, they are often located in traffic arteries or crowded places, such as tourist attractions, where there are usually many buildings and a variety of public facilities, e.g., telecommunications and power towers and power lines, which may surround or cross the entire area affected by a given disaster. Therefore, in

order to fully adapt to the complex environments in which geo-hazards may be located, UAVs used to carry out emergency investigation should meet some basic requirements, e.g., small size, light weight and quick assembly and disassembly; easy takeoff and landing and a lack of special site requirements; simple, flexible and convenient flight control and image collection; a stable flight control system and a reliable failure protection function; strong wind resistance and a reliable aerial gimbal system for carrying the camera; a powerful ground control station and a stable image and data transmission system;

and a certain endurance that guarantees that a flight can cover the entire area affected by most single geo-hazards.

According to the above requirements, combined with the comprehensive considerations of applicability, security, stability and economy (which mainly refers to the low cost of initial construction and later maintenance of the UAV system), a number of on-site tests and practical applications were carried out for single landslides in the Three Gorges Reservoir area, China. Finally, a UAV system was customized. A photograph of this system is shown in Fig. 1, the system architecture and

the main function modules are shown in Fig. 2, and the core components and main features of the customized UAV are shown in Table 1.

The customized UAV system consists of four subsystems, including a multi-rotor UAV subsystem, an aerial photography subsystem, a ground control subsystem and a ground surveillance subsystem, which are described in detail below.

(1) A customized four-axis and eight-rotor carbon fibre airframe is used. The outstanding advantages of this design include

its high strength and power and light weight (the whole aircraft with its camera and the aerial gimbal system is less than 5 kg). In addition, compared with fixed wing aircraft, it is smaller, has better manoeuvrability and enhanced fixed-point hovering capability. In particular, special sites for taking off and landing are not required, which is very important for using UAVs to quickly investigate single geo-hazards, which are usually located in complex environments. Although the battery life is poor (it provides approximately 20 minutes of flight time), if an average flight speed of 5 m/s is considered, the resulting flight distance of approximately 5 km guarantees that one flight is sufficient to cover the whole area of most single geo-hazards. Even if multiple flights are required, only the battery needs to be replaced.

(2) The flight control system, which directly controls the processes of flight and image collection, is the "brain" of the UAV system, and its performance and stability directly determine the functioning and the security of the whole system. Currently, many mature commercial flight control systems exist; however, they are closed-source systems. Thus, in the event of failure, the unit must be returned to the factory for repair or re-purchased, resulting in high costs in money and time. Therefore, a widely used open source flight control system, i.e., Pixhawk 2.4.5 (Meier et al., 2012) that has dual processors, is used. The high robustness and powerful data processing capacity of this unit have been recognized. Equipped with a high-performance Global Navigation Satellite System (GNSS) module, data and image transmission modules, etc., this flight control system provides complete support for several necessary functions, such as route planning, flight positioning, real-time data and image transmission, and so on.

(3) The aerial photography subsystem, which is used to collect overhead or oblique high-resolution images of geo-hazards, serves as the core data source of the emergency investigation information. Thanks to the rapid development of tools for semi-automated photogrammetric processing, especially Structure from Motion (SfM) photogrammetry (Westoby et al., 2012; Li et al., 2013; Förstner and Wrobel, 2016; Özyeşil et al., 2017), which is based on computer vision algorithms, the requirements of early aerial photography equipment have been greatly reduced. Therefore, only a Sony HX200 digital camera with 18 effective megapixels and a Vario Sonnar T* 4.8-144 mm F/2.8–5.6 lens is used to take the photographs. In addition, in order to keep the camera stable to ensure the collection of clear images or accurately adjust the lens orientation according to actual needs, a three-axis brushless aerial gimbal system is used to carry the camera. The camera shutter and the aerial gimbal system are entirely controlled by the flight control system.

(4) The ground control subsystem mainly includes the ground control station, which is a notebook computer equipped with the UAV's ground control software. For compatibility with the Pixhawk flight control system, the corresponding open source ground station software, i.e., Mission Planner 1.3.39 (Team, 2016) is used. In addition, by interacting with the flight control system, the software can carry out several core functions, including the debugging and maintenance of the UAV, setting parameters, and route planning, monitoring and control. In a word, using the ground control station and the flight control system, the whole process of flight and image collection can be fully automated. In addition, a remote control provides manual control of the flight in case of emergency.

(5) The ground surveillance subsystem should be established to display real-time flight images and flight state data. This subsystem provides timely and accurate information that enables operators to make effective judgements, decisions, and manipulations, to ensure the flight safety. In the key on-screen-display (OSD) module, the flight state data are superimposed on the real-time flight images. In addition, then, all of this information is transmitted to the ground terminal monitor using

the image transmission system (including the image delivery module on the aircraft and the image receiving module on the ground terminal monitor).

The customized UAV system (including the Sony HX200 digital camera, but not including the computer that is equipped with the ground control software) costs only $1825, which is equal to the price of the DJI Phantom 4 pro, a current and very popular consumer-level UAV. However, the UAVs are usually used to take wide-angle photos, so with the same aperture (f 2.8), the shorter 4.8 mm focal length lens of Sony HX200 can capture more clear photos than the DJI Phantom 4 pro with a 8.8 mm focal length lens, because the latter has a more serious background blur. Moreover, due the stronger power and greater weight, the customized UAV displays more flexible control and better wind resistance. In short, our customized UAV system has carried out more than 20 emergency investigations of single geo-hazards in the Three Gorges Reservoir area. Satisfactory photographs of every geo-hazard have been obtained, and there has not been a runaway accident, which demonstrates that the system has substantial applicability, safety and reliability and is very suitable for the emergency investigation of single geo-hazards, even in mountainous environments like that of the Three Gorges Reservoir area.

## 3 Implementation process

The implementation process of using a UAV to perform emergency investigations of single geo-hazards can be divided into four steps, i.e., indoor preparation, site investigation, on-site fast processing and application, and indoor comprehensive processing and application. Fig. 3 shows this process in detail and the tasks involved in each step, which are described below.

### 3.1 Indoor preparation

Performing the necessary indoor preparation can improve the efficiency of on-site emergency investigation and mainly includes battery charging, initial inspection of the UAV system, and preliminary route planning.

### 3.1.1 Battery charging

At present, our system components use lithium batteries. To protect the efficiency and prolong the service life of lithium batteries, when they are not in use, the voltage of the lithium cells should be maintained at 3.8 v or so (Broussely, 2002), that is, neither fully charged nor empty. Therefore, charging of all of the lithium batteries used by every component, including the UAV, the camera, the remote control, the notebook computer equipped with the ground control station, and the terminal monitor, is the primary indoor task.

### 3.1.2 Initial inspection of the UAV system

The primary purpose of the initial inspection of the UAV system is to avoid failures in the core components that cannot be restored quickly during the site investigation process. In addition, this inspection establishes whether the main components, e.g., the flight control system, propellers, GNSS and compass, data and image transmission module, aerial gimbal system and camera, ground control station, terminal monitor, and remote control are working properly.

### 3.1.3 Preliminary route planning

In addition, if the location of a geo-hazard can be determined, it is necessary to carry out preliminary route planning indoors based on publicly available satellite maps, e.g., Google Earth, Bing Maps, AutoNavi Maps, etc., in the Mission Planner software package. Typically, the preliminary flight routes are simply designed as a regular grid pattern in plane view that

175 covers the whole area affected by the geo-hazard (Fig. 4). In a word, preliminary route planning can help to save time spent on detailed route planning on-site. Of course, if the location cannot be determined in advance, the indoor preliminary route planning cannot be performed, but it does not affect the subsequent on-site investigation using the UAV.

## 3.2 On-site investigation

Without a doubt, the on-site investigation is the most important step. We believe that there is an important principle to be
180 followed; that is, the ultimate goal is fast and efficient collection of high quality photographs of the geo-hazard for subsequent processing and application, but this collection must meet safety requirements. That is, the safety of all on-site personnel, buildings and public facilities must be ensured, and the safety of the UAV system must be taken into consideration.

### 3.2.1 Environmental assessment

185 Before commencing a formal on-site investigation, an environmental assessment is required to determine the UAV-based investigation scheme. Usually, an assessment of the environment surrounding the geo-hazard and its characteristics, as well as the implementation conditions are needed. The former includes the local topography and meteorological conditions, the distribution of aerial and ground facilities, visual range and intervisibility, flight range and other judgements. Assessment of the geo-hazard characteristics includes the topography of the area affected by the hazard; its length, width, area, plane shape,
190 and elevation change; and the risk it poses. In addition, assessment of the implementation conditions includes the number of GNSS satellites, the strength and stability of the signal, the stability of the electronic compass, the layout of ground control points (GCPs) and the location used for takeoff and landing.

### 3.2.2 Two UAV-based investigation schemes

Based on a number of previous investigations, two investigation schemes, including automatic and manual investigation, are
195 described as follows.

### (1) Automatic scheme

Using the automatic scheme, the UAV system is capable of autonomous flight in accordance with the routes established during detailed planning, as well as automatic photo image collection, using the UAV's own GNSS, compass and barometer. This scheme requires no manual intervention under normal conditions and is therefore safer and more reliable than manual
methods. At the same time, the acquisition of high quality photographs is more likely in this mode, which can automatically meet some requirements of the subsequent photogrammetric processing, e.g., the frontal and side overlap ratios between photographs. In view of this, as long as there are more than five stable GNSS satellite signals in the geo-hazard area, the vast majority of emergency investigations should use the automatic investigation scheme. In addition, this scheme is divided into six steps (Fig. 3), which are described below.

● **Layout and measurement of GCPs**

To improve the accuracy of the photogrammetric processing results, the establishment and measurement of GCPs in the field is essential (Niethammer et al., 2012; Lucieer et al., 2014; Niu et al., 2014). Usually, three to five GCPs should be established in or around the geo-hazard (see section 4.1 for details). Real-time kinematic (RTK) differential Global

Positioning System (DGPS) techniques, which have advantages including speed, high efficiency and high precision, should be used to measure the 3D coordinates of all GCPs.

● **Assembly of the UAV system**

A modular design is used in our customized UAV system. The transport of disassembled components saves space, and the components are also protected from squeezing or crushing during the transport process. Therefore, after arriving at the disaster site, the modules need to be quickly assembled to form the complete UAV system.

● **Full inspection of the UAV system**

After the system has been assembled, all of the subsystems need to be fully checked with the power turned on. The main purpose is to eliminate hidden dangers on the ground and to ensure flight safety and normal photo image collection. This step is very important and cannot be ignored.

● **Detailed route planning**

Automatic investigations must rely on detailed route planning. If preliminary route planning has been carried out indoors (section 3.1.3), the detailed route planning should be based on this preliminary plan. Otherwise, detailed route planning should be carried out at the site. The core of this step is the determination of the route types according to the characteristics of the geo-hazard, as well as accurate establishment of the waypoint positions and the actions of the UAV, the aerial gimbal system, and the camera. See section 4.2 for details.

● **Setting of parameters**

The setting of parameters is the last step before flight, and it cannot be neglected. Several important control parameters must be set according to the actual scene. Typically, the recommended flight rate is 5 to 20 metres per second, and the camera image collection rate should not be less than 1 picture per second. It is important to remember to import all of the parameters into the flight control system onboard the UAV from the ground control station. These parameters can then take effect.

● **Autonomous flight and automatic photo image collection**

A relatively flat and open place should be selected as the takeoff and landing site. After taking off, the UAV should follow the planned route for autonomous flight and automatic photo image collection under normal circumstances. During the flight, the status of the UAV and camera should be closely monitored (see section 4.3 for details). In the event of an abnormal state, the UAV should be switched to the manual mode to permit emergency response. After the flight is completed, the UAV system and the quality of the photographs should be checked immediately.

**(2) Manual scheme**

In the manual scheme, the entire UAV flight and the process of photo image collection has to be manually controlled using the remote control. This scheme requires no route planning, which can save time in site investigation. However, it requires excellent piloting skills, and the flight safety and image quality are frequently degraded; accordingly, the flight process should be monitored intensively. Therefore, use of the manual scheme should be avoided, although, in some places, such as mountainous areas or canyons, where the GNSS signals are unstable or even absent, or the scope of the geo-hazard is extremely limited, the manual scheme may be more suitable. In addition, this scheme is divided into 4 steps (Fig. 3), which are briefly described below.

- **Layout and measurement of GCPs**

If there is no GNSS signal, the captured photographs will not be associated with GNSS locations. In this case, the layout and measurement of GCPs is indispensable in supporting the subsequent photogrammetric processing. The setting of GCPs is the same as in the automatic scheme (see section 4.1 for details). However, the use of a total station is recommended for measuring the GCPs where GNSS signals are absent.

- **Assembly of the UAV system**

This step is the same as in the automatic scheme.

- **Full inspection of the UAV system**

This step is the same as in the automatic scheme.

- **Manual flight and photo image collection**

Compared with the automatic scheme, the system status and the quality of the photographs should be monitored more carefully during the flight (see section 4.3 for details). In addition, it is more important to check the system and the photographs after the flight. In particular, the quality, scope and overlap rate of the photographs must be evaluated.

## 3.3 On-site fast processing and application

After on-site UAV-based investigation is completed, the low-resolution photographs can be subjected to fast photogrammetric processing using a portable computer in the field. In general, in only ten to several tens of minutes, some rough results with approximately metre-level accuracy can be generated, including digital surface models (DSMs), digital orthophotos, and three-dimensional models. Fast processing focuses on the speed with which the results are generated, not their precision. Although the accuracy is relatively poor, these emergency investigation results that can be obtained quickly in the field still provide important support for the rapid on-site development of preliminary emergency response plans for geo-hazards. This high speed is the most prominent advantage of the UAV-based method for emergency investigation of single geo-hazards, compared with traditional methods. This processing is divided into 4 steps (Fig. 3).

- **Preprocessing of photographs**

Preprocessing of photographs includes selecting photos that cover the extent of the geo-hazard, removing poor-quality photographs (e.g., blurred images), and checking the GNSS information associated with the photographs. In general, photographs taken using the manual scheme require more time for pretreatment than those collected using the automatic scheme.

- **Fast SfM processing and generation of coarse-precision results**

Compared with traditional digital photogrammetry method, the semi-automated SfM-based photogrammetric method is recommended for use in the processing of UAV-based photographs because it is simpler and more efficient (Snavely, 2008; Westoby et al., 2012; James et al., 2016); for example, the camera position can be automatically calculated using only the GNSS data associated with each photograph, and information on the attitude of the aircraft, such as its roll, pitch, and yaw, obtained from the inertial measurement unit (IMU) are no longer needed (Huang et al., 2017). The adopted SfM-based photogrammetric processing consists of extracting tie points from the photographs at lower resolution, performing GPS-assisted aerial triangulation and bundle adjustment, and generating three-dimensional point clouds. Using the dense

point cloud, coarse-precision results for the geo-hazard, including a DSM, a digital orthophoto and a three-dimensional
model, can be generated. The Pix4Dmapper software package (Strecha et al., 2012; Mesas-Carrascosa et al., 2015) is used to
process the photographs by SfM photogrammetric methods. To enable fast processing, the lower image scale is set first, and
only then is the GNSS information associated with each photograph used during the aerial triangulation and bundle
adjustment. That is, the coordinates of the GCPs are not introduced during this stage to improve the absolute spatial position
accuracy. Because the M8N GPS module with a nominal positioning accuracy of 2.5 m was used in our aircraft, the fast SfM
processing results generally displayed coarse precision and metre-level error.

● **Coarse quantification and display of the geo-hazard**

Based on the coarse-precision results for the geo-hazard, using geographic information system (GIS) or remote sensing (RS)
software, the basic characteristics of the geo-hazard can be quantified. These characteristics include length, width, area, and
elevation change. In addition, the three-dimensional scene of the geo-hazard and its surroundings can be vividly displayed.

● **Supporting the development of the preliminary emergency response plan**

The quantitative characteristics and the intuitive three-dimensional scene of the geo-hazard provide the basis and macro-level
information for the rapid on-site development of a preliminary emergency response plan. The metre-level error of the results
essentially does not affect the appropriateness of such qualitative plans.

## 3.4 Indoor comprehensive processing and application

The design of detailed emergency response plans is an important step in the implementation of disaster prevention and
mitigation efforts, so the basic data such as terrain representations and orthophotos that are used in the design must be
accurate and clear. The purpose of comprehensive processing is to obtain such high quality results. Therefore, the original
photographs are reprocessed using high-performance desktop computers or graphic workstations indoors. The
comprehensive processing generally takes one to several hours, but all of the results have centimetre-level accuracy because
the GCPs are introduced. Comprehensive processing focus on the precision of the results, rather than the speed with which
they are generated. It is divided into 3 steps (Fig. 3).

● **Comprehensive SfM processing and generation of high-precision results**

The comprehensive SfM processing workflow is the same as that used in the fast processing. The differences are that the
original photographs with high resolution are used, and the GCPs are introduced before the point clouds are generated.
Accordingly, the products of the comprehensive SfM processing are the same as those of the fast processing. That is, DSMs,
digital orthophotos and three-dimensional models are produced, but these products are high-precision and high-resolution.
Likewise, Pix4Dmapper software is used for the comprehensive SfM processing. First, the full-scale photographs with
GNSS information are used. Subsequently, during the process of aerial triangulation and bundle adjustment, the GCPs are
introduced to improve the absolute spatial position accuracy. Because the 3D coordinates of the GCPs are measured using
the RTK-DGPS technique and have a nominal positioning accuracy of 2 cm, the comprehensive SfM processing results were
generally high-precision with centimetre-level error.

● **Accurate quantification and display of geo-hazards**

Using the high-precision and high-resolution results for the geo-hazard, the basic characteristics of the geo-hazard can be

accurately quantified. Accordingly, the three-dimensional scene of the geo-hazard and its surroundings can be more accurately and vividly displayed.

● **Supporting the design of detailed emergency response plans**

Based on the accurate quantitative characteristics, a large-scale topographic map and plan can be produced, and accurate design data can be obtained from the high-precision and high-resolution DSM, orthophoto, and three-dimensional scene of the geo-hazard. This information provides important support for the design of detailed emergency response plans.

## 4 Key techniques and methods

### 4.1 Layout and measurement of GCPs

Due to the limited precision of the GNSS units carried by UAVs (e.g., the M8N GPS module onboard our aircraft has a nominal positioning accuracy of 2.5 m), it is necessary to set and measure GCPs in the field before the UAV flight and image collection, to improve the accuracy of the photogrammetric processing results. In addition, the layout and measurement of the GCPs should be implemented quickly and efficiently, but the results should be high-precision.

First, within the area covered by the flight, some obvious ground feature points, e.g., house corners, road intersections, exposed bedrock, etc., can be used directly as GCPs, as long as they can be clearly identified both on the ground and on photographs. Otherwise, several GCP markers that can also be identified in photographs need to be placed on the ground. Usually, for single geo-hazards, in order to balance requirements in terms of accuracy and efficiency, according to our practical experience, only three to five GCPs need to be established in or around the geo-hazard, and the distribution should be as uniform as possible, e.g., networks made up of equilateral triangles or quadrilaterals are appropriate (Ai et al., 2015; Pix4D, 2017). It is worth noting that the layout of the GCPs should be completed before the UAV flight and image collection, to ensure that the photographs contain all of the GCPs. Regarding the measurement of the GCPs, the RTK-DGPS technique, which has advantages in that it is fast and has high efficiency and high precision, should be used preferentially as long as there are stable GNSS signals, regardless of whether the automatic or manual scheme is used, to measure the 3D coordinates of all of the GCPs. On the other hand, in mountainous areas, canyons, etc., with unstable or even no GNSS signals, the total station measurement techniques would be a good choice, and sometimes even the non-prism total station measurement techniques may be the only option (Huang et al., 2017). Moreover, the measurement can be carried out at any time during the on-site investigation process, but if it is performed at the same time as the collection of images by the UAV, the GCP markers should not be covered.

### 4.2 Route planning

According to the characteristics of individual geo-hazards, proper route type selection and accurate motion design are key in ensuring the safety and efficiency of UAV-based emergency investigation. Based on a number of examples, three typical route types are summarized as follows (Fig. 5).

(1) Planar grid pattern for slightly inclined slopes (Fig. 5a). This pattern is suitable for geo-hazards that cover large areas (typically several million square metres) on the gentle slopes (typically less than 40 °), such as gently inclined landslide bodies. The primary purpose of the emergency investigation for this kind of disaster is to obtain a digital terrain model and

an orthophoto. Therefore, the planned route consists of a regular planar grid that covers the whole planar area of the geo-hazard. In addition, the camera lens always points vertically down to the ground (i.e., the lens orientation is held at 0 °).

It is worth noting that the flying height of the route should be dynamically adjusted to meet the elevation changes of the disaster and slope. In principle, it is advisable to maintain the UAV's flying height at a constant distance (i.e., the $h$ in Fig. 5a) from the ground, and practice shows that $h$ in 50 m ~ 100 m is proper. Lower flights require longer routes and increased flight time, and flight safety decreases; conversely, higher flights increase the mean ground sampling distance (GSD) of the acquired images, thus providing for lower spatial resolution of the final photogrammetric products.

(2) Vertical grid pattern for steep slopes (Fig. 5b): This pattern is suitable for geo-hazards that are developed on steep slopes (typically greater than 60 °), such as dangerous rock masses on cliffs. Emergency investigations of this kind of disaster should aim at obtaining orthophotos of the facade and 3D models, rather than digital terrain and vertically downward orthophotos. In this case, the planned route consists of a regular vertical grid which that covers the whole facade area of the geo-hazard. In addition, the camera lens always points horizontally to the disaster body (i.e., lens orientation is held at 90 °).

The plane positions of all of the horizontal routes can overlap, but they are at different altitudes. In addition, it is advisable to keep the UAV flying at a constant distance (i.e., the $d$ in Fig. 5a) from the area affected by the disaster (practice shows that a $d$ of 40 m ~ 80 m is proper).

(3) Combined grid pattern for transitional terrain (Fig. 5c): This pattern is suitable for geo-hazards that are located on transitional terrain and include both gentle and steep slopes, such as a dangerous rock mass on a cliff and the corresponding

collapse accumulation mass on the gentle slope below. The main purposes of emergency investigation for this kind of disaster are to obtain a digital terrain model and an orthophoto, as well as a facade orthophoto and a 3D model. Therefore, the combined grid pattern should be adopted for the planned route. That is, a regular planar grid is used to cover areas with gentle slopes, and a vertical grid is used to cover areas with steep slopes. Accordingly, the camera lens points vertically down to the ground within the part with the planar grid (i.e., the lens orientation remains at 0 °), and gradually rises from the low

position to the high position in the part with the vertical grid (i.e., the lens orientation changes from 0 ° to 90 °). The flying height $h$ and flying distance $d$ in Fig. 5c can be set as in the planar and vertical grid patterns, respectively.

In particular applications, the planned route should be selected from the three typical route types listed above; alternatively, these routes can be changed flexibly or combined, based on the spatial distribution characteristics of the specific geo-hazard being investigated. However, in any case, the planned route must meet the requirements that the obtained pictures' frontal

overlap ratios must be at least 75%, and the side overlap ratios must be at least 60%. Otherwise, the scope and accuracy of the post-processing results will be seriously affected.

In addition, the detailed route planning in the field should also account for the following points.

①Whether a preliminary route planning has been carried out or not, it is necessary to accurately calibrate the flight route and range based on the actual locations from the UAV's own GNSS data.

②The route coverage should be larger than the actual area affected by the geo-hazard to ensure that the photographs of the disaster overlap sufficiently.

③The starting point and route should be established near the foot of the disaster body, and the end of the route and the ending point should be set near the top. Thus, the altitude of the UAV will progress from low to high altitude during the

emergency investigation (Fig. 5). The UAV is more stable during upward flight, which is more conducive to taking clear photographs.

④After careful checking, the planned route must be imported into the flight control system of the UAV to take effect.

## 4.3 Flight and image collection process control

It is essential to carry out the pre-flight inspection after importing the accurate planned route data and setting the flight parameters. This inspection mainly includes assessments of the battery capacity, GNSS signal, propeller, aerial gimbal system, camera, data and image transmission modules, remote control and the ground control station. The UAV can then be used to take photographs for the emergency investigation of single geo-hazards. During flights, it is best to have three technical staff involved in the implementation to ensure the flight safety and photo quality. The primary operator, in the automatic scheme, is responsible for monitoring the flight and image collection state through the ground control station during the normal autonomous process or switching to manual operation of the flight and photo image collection in the event of an abnormal state. When the manual scheme is in use, the primary operator is always responsible for manually performing the taking off, flight and landing of the UAV using the remote control. The primary supervisor is always responsible for monitoring the real-time flight images and changes to the important parameters (e.g., the height, the speed, the battery capacity, and the GNSS signal) through the ground terminal monitor. The primary supervisor immediately notifies the primary operator of changes in the UAV's flying state, regardless of whether the automatic or manual scheme is in use. Meanwhile, the deputy operator and monitoring personnel, in the two schemes, is responsible for real-time tracking of the posture changes of the UAV and observing the surroundings ahead of the UAV through a spotting scope, to detect the aircraft anomalies or flight obstacles as early as possible, and promptly notifies the primary operator for emergency response; in the manual scheme, this staff member is also responsible for manipulating the camera lens and image collection using another remote control.

## 4.4 SfM photogrammetric processing

At present, traditional digital photogrammetry and the newly developed SfM photogrammetric method, which is based on computer vision algorithms, can both be used for the processing of UAV images, but the latter is simpler and more efficient. In contrast to traditional photogrammetric methods, which require a single stereo pair of images in addition to the 3D locations and orientations of the cameras or the 3D locations of a series of GCPs, the SfM technique requires only multiple overlapping photographs as input (Westoby et al., 2012). The principles and workflow of SfM have been described by Snavely (2008), Snavely et al. (2008), and Westoby et al. (2012). The Pix4Dmapper software package is used for the SfM photogrammetric processing, which can convert a large number of images into georeferenced 2.5D DSMs, digital orthophotos and 3D models (Huang et al., 2017).

When the SfM photogrammetric processing method is used to process the photographs that are captured by UAVs during the emergency investigation of single geo-hazards, it is divided into on-site fast processing and indoor comprehensive processing (Fig. 3). In addition, the results of the SfM photogrammetric processing should also be targeted to the different types or characteristics of the geo-hazard; e.g., for the type of event shown in Fig. 5a, the main results should be a digital terrain model and an orthophoto; for the type of event shown in Fig. 5b, the core results will likely be a facade orthophoto

and a 3D model; for the type of event shown in Fig. 3c, the results should include a digital terrain and orthophoto, as well as a facade orthophoto and a 3D model.

## 5 Application examples

### 5.1 Emergency investigation of a slightly inclined landslide

In early September 2014, a mass movement occurred under the influence of continuous heavy rainfall and resulted in a landslide in the Three Gorges Reservoir area. This landslide represented a serious threat to the surrounding houses, highway traffic and the life and property of the local residents (Fig. 6a). Environmental assessment showed that the landslide had a gentle slope and small size, but it threatened a large area. In addition, the environment was rather open and the GNSS signal was stable. Therefore, the automatic investigation scheme was adopted.

First, a route was planned according to the pattern of Fig. 5a based on the position of the mass movement and its extent. At the same time, 4 GCPs were selected around the landslide (Fig. 6a), and the RTK-DGPS with a nominal positioning accuracy of 2 cm was used to measure the 3D coordinates of these GCPs. The establishment and measurement of the GCPs took approximately 50 minutes. Then, by autonomous flight and automatic photo image collection, 66 photographs were captured, and the mean GSD of the original images was 3.67 cm/pixel. Including route planning and UAV preparation, the entire working time using the UAV required only approximately 30 minutes. Finally, using SfM photogrammetric processing, an orthophoto with a spatial resolution of 4.25 cm (Fig. 6a) and a 3D texture model (Fig. 6b) were generated. Usually, a rigorous accuracy assessment should be performed by using external and independent check points, but for simplicity and efficiency, the 4 GCPs were used also as the checkpoints for accuracy assessment, and the mean, standard deviation (*Sigma*), and root-mean square error (*RMSE*) values were calculated. The results showed that the fast SfM processing required only 28 minutes, but the planar errors were about 2.6 m ($Error\_X_{mean}$ = 2.327 m, $Error\_Y_{mean}$ = 2.862 m; $Error\_X_{sigma}$ = 0.166 m, $Error\_Y_{sigma}$ = 0.114 m; $Error\_X_{RMSE}$ = 2.331 m, $Error\_Y_{RMSE}$ = 2.864 m), and the vertical error was more than 4.0 m ($Error\_Z_{mean}$ = 4.165 m, $Error\_Z_{sigma}$ = 0.211 m, $Error\_Y_{RMSE}$ = 4.169 m). In contrast, the comprehensive processing took 65 minutes, and because the GCPs were introduced during the SfM photogrammetric processing, the planar errors were reduced to about 0.035 m ($Error\_X_{mean}$ = 0.031 m, $Error\_Y_{mean}$ =0.038 m; $Error\_X_{sigma}$ = 0.009 m, $Error\_Y_{sigma}$ = 0.014 m; $Error\_X_{RMSE}$ = 0.032 m, $Error\_Y_{RMSE}$ = 0.04 m), and the vertical error was less than 0.05 m ($Error\_Z_{mean}$ = 0.045 m, $Error\_Z_{sigma}$ = 0.012 m, $Error\_Y_{RMSE}$ = 0.046 m). In short, all of the on-site work, including the UAV-based investigation, the layout and measurement of the GCPs, and the fast photogrammetric processing, required a total of 78 minutes. In addition, an orthophoto and a 3D texture model with an error of approximately 3 m were generated on-site. These products provided macro-level information about the landslide and its surroundings.

Based on the results of the above emergency investigation, combined with ground investigations, the characteristics and effects of the landslide were quickly interpreted (Fig. 6c) and evaluated. These products revealed the formation of obvious head and side scarps. Moreover, the drainage ditch that was located within the landslide was completely destroyed, and the front loose soil mass collapsed and blocked the gully, forming a free face and leading to the emergence of a large number of tension cracks. All of these indications suggested that the landslide represented an obvious mass movement, and that the landslide was in an unstable state. In addition, the landslide was a direct threat to the houses which were located adjacent to

the right boundary of the landslide. Moreover, as the loose soil mass at the front of the landslide continuously accumulated in
the gully, a debris flow would be easily triggered by heavy rainfall, then seriously threatened the house and highway. Based
on the conclusions above, emergency response measures were put forward, including using professional monitoring
techniques such as GNSS, extensometers, rain gauges, and ground-based inspection to continuously track the process of
deformation and induction of the landslide; establishing a citizen science-based monitoring and prevention system, which
means to encourage the surrounding population to watch for signs of deformation within the landslide body, e.g., cracks,
collapses, and cracks in houses, especially in association with heavy or continuous rainfall; and developing an emergency
evacuation programme to ensure the orderly avoidance and reduction of losses before additional movement occurs.

This application shows that the results of UAV-based emergency investigation can provide a large-scale perspective for
use in the comprehensive evaluation of the characteristics of single geo-hazards and their potential impacts, which can make
up for the defect of ground-based investigations, which focus on parts of geo-hazards but ignore the whole.

**5.2 Emergency investigation of a dangerous rock mass on a steep cliff**

In September 2015, a dangerous rock mass was noted above a provincial highway on the left bank of the Yangtze River in
the Three Gorges area. This rock mass presented a serious threat to the safety of the highway traffic and shipping along the
Yangtze River (Fig. 7a). Only one side of the dangerous rock mass was attached to a steep cliff (Fig. 7b), and it was located
at least 100 m away from the lower highway, which caused the site to be inaccessible to human beings. Therefore, the use of
470 UAVs for emergency investigation was prioritized. Because a stable GNSS signal existed at the study site, the automatic
investigation scheme was adopted.

First, route planning was performed according to the pattern shown in Fig. 5b. At the same time, 3 GCPs were established
along the highway and measured with the RTK-DGPS technique (Fig. 7c). In addition, it should be noted that, because the
three GCPs were located in nearly a straight line, given the limited environment, which could not meet the processing
requirements, another GCP was established on top of a hill behind the cliff (Fig. 7c), and its coordinates were measured
according to the pre-existing topographic map. The layout and measurement of the 3 GCPs took only approximately 25
minutes. The camera lens direction was then set at approximately 45 ° to the steep cliff. Through autonomous flight and
automatic photo image collection, 104 photographs were captured, and the mean GSD of the original images was 4.92
cm/pixel. Together with route planning and UAV preparation, the entire working time using the UAV took 35 minutes.
Finally, in order to obtain high-precision results for the design of the detailed emergency response plan, the comprehensive
SfM processing was used directly, which took approximately 100 minutes, and a DSM and a 3D texture model (Fig. 7c, d)
with a spatial resolution of 5.47 cm were generated. Similarly, the 4 GCPs were used as checkpoints for accuracy assessment.
The results showed that the spatial errors were 0.237 m ($Error\_X_{mean}$ = 0.218 m, $Error\_Y_{mean}$ = 0.183 m, $Error\_Z_{mean}$ = 0.310
m; $Error\_X_{sigma}$ = 0.348 m, $Error\_Y_{sigma}$ = 0.292 m, $Error\_Z_{sigma}$ = 0.410 m; $Error\_X_{RMSE}$ = 0.372 m, $Error\_Y_{RMSE}$ = 0.312 m,
$Error\_Z_{RMSE}$ = 0.471 m), i.e., the accuracy was sub-meter-level, primarily because the GCP on the hilltop could not be
accurately measured. Even so, the sub-meter level accuracy was able to meet the requirements of quantifying the size and
generating large-scale topographic maps of the dangerous rock mass for use in the design of emergency control measures and
would not have a substantial impact on the conclusions. Moreover, all of the investigative work, extending from the on-site

UAV-based investigation to the generation of high-precision results, required only approximately 2 hours and 15 minutes. Such a short time period could not be achieved through ground-based investigation, especially for an isolated dangerous rock mass on a cliff.

In view of the DSM and the 3D model, the results showed that the dangerous rock mass was 24 m high, 12 m wide, and 12 m thick and had a volume of 3456 m$^3$, and its exact distance from the highway was 110 m. The lower and upper part of the rock mass had fallen, so it was completely unsupported. The left and right boundary cracks were fully connected. All of these signs indicated that the dangerous rock mass was in an unstable state. Therefore, the relevant emergency investigation results described above were submitted to the technical department to support the design of a detailed emergency response plan.

In this case, the UAV provided the only reasonable means of performing an emergency investigation of this dangerous rock mass on a steep cliff, and the DSM and the 3D texture model provided both full and partial information that supported the design of the emergency response.

## 5.3 Emergency investigation of a combined slope on transitional terrain

In January 2016, some rockfalls occurred on a high and steep artificial slope that had been controlled in the Three Gorges Reservoir area, which presented a serious threat to the safety of the traffic on a provincial highway and the shipping on the Yangtze River below (Fig. 8a). The slope was located on the left bank of a tributary estuary of the Yangtze River, and it was surrounded by the rivers on three sides. Geomorphologically, the slope above the highway consisted of five steep cliffs (Fig. 8b), whereas the portion of the slope below the highway has a gentle slope but is surrounded by the rivers. It could be seen that it was almost impossible to implement a ground-based investigation to identify all of the potential geo-hazards, so the UAV was used and the automatic investigation scheme was adopted.

First, route planning was performed based on the pattern shown in Fig. 5c. At the same time, four GCPs were arranged along the winding highway and measured with the RTK-DGPS technique (Fig. 8a). The layout and measurement of the GCPs required approximately 40 minutes. Using autonomous flight and automatic photo image collection, 75 photographs were then captured, and the mean GSD of the original images was 4.51 cm/pixel. Together with route planning and UAV preparation, the entire working time using the UAV took only approximately 50 minutes. Finally, using SfM photogrammetric processing, an orthophoto, a DSM and a 3D texture model with a spatial resolution of 5.02 cm were generated. The 4 GCPs were used as checkpoints for accuracy assessment. The results showed that the fast SfM processing took 40 minutes and the spatial errors were 3.686 m ($Error\_X_{mean}$ = 3.173 m, $Error\_Y_{mean}$ = 3.401 m, $Error\_Z_{mean}$ = 4.485 m; $Error\_X_{sigma}$ = 0.505 m, $Error\_Y_{sigma}$ = 0.409 m, $Error\_Z_{sigma}$ = 0.411 m; $Error\_X_{RMSE}$ = 3.203 m, $Error\_Y_{RMSE}$ = 3.419 m, $Error\_Z_{RMSE}$ = 4.499 m). In contrast, the comprehensive processing took 95 minutes, and the spatial errors were reduced to 0.061 m ($Error\_X_{mean}$ = 0.053 m, $Error\_Y_{mean}$ = 0.060 m, $Error\_Z_{mean}$ = 0.069 m; $Error\_X_{sigma}$ = 0.016 m, $Error\_Y_{sigma}$ = 0.019 m, $Error\_Z_{sigma}$ = 0.028 m; $Error\_X_{RMSE}$ = 0.054 m, $Error\_Y_{RMSE}$ = 0.062 m, $Error\_Z_{RMSE}$ = 0.073 m) by introducing the GCPs. In short, all of the on-site work required a total of 90 minutes.

Based on all of the results, mainly the high-resolution 3D texture model, three potential geo-hazards were identified on this slope (Fig. 8). Above the highway, within the 4th section of the cliff and the left part of the 5th section of the cliff, several dangerous isolated rock masses had formed because of the continuous development of tension cracks. In the upper

part of the 2nd section of the cliff and the left part of the 3rd section of the cliff, a large number of broken rock masses that could fall easily had developed due to the continuous development of two sets of tension cracks. Below the highway, a tension crack existed on the right side of the slope. In addition, the detailed characteristics of each section of the cliff could be accurately measured (Fig. 8b). The relevant results from the emergency investigation were submitted to the relevant departments for risk assessment and the design of control measures.

In this case, the flexibility and wide applicability of the UAV were fully proven. UAV-based methods can provide high-resolution visual 3D scenes and models, as well as accurate quantitative basic terrain data, thus providing strong support in the evaluation of single geo-hazards or the design of relevant control measures.

According to the application examples described above, it can be seen that UAV-based investigations can be completed on-site within 1 hour from Table 2. And if needed, coarse-precision results with metre-level error can also be generated on-site using fast SfM processing, usually within 1 hour. That is, using UAV-based emergency investigation methods and SfM photogrammetric processing technology, macro-scale and three-dimensional information on a single geo-hazard can be obtained within 2 hours, which can support the rapid on-site development of preliminary emergency response plans. This rapidity is the most prominent advantage of this UAV-based method in comparison with the traditional methods. Moreover, by introducing the GCPs into the comprehensive SfM processing, high-precision results with centimetre-level error can also be obtained. These high-precision results can support the design of detailed emergency response plans, and the processing time required was typically several hours.

## 6 Conclusions

This paper comprehensively describes the method of using UAVs for emergency investigation of single geo-hazards. The main conclusions are summarized below.

(1) According to the requirements of emergency investigation, combined with comprehensive consideration of applicability, security, stability and economy, the UAV system used is custom-built. Its core functions and modules include a four-axis and eight-rotor carbon fibre airframe; a stable, reliable, and open source flight control hardware system that is compatible with the ground control station software, which provides comprehensive support for route planning, autonomous flight and automatic photo image collection; an ordinary digital camera with a relatively high number of pixels or a single-lens reflex (SLR) camera, which is satisfactory for image collection (a three-axis brushless aerial gimbal system is used to ensure the collection of clear images and the flexible adjustment of the camera lens direction); and a ground surveillance subsystem, which is used to monitor the flight of the UAV system and the collection of images by this system.

(2) The process of using the UAV to perform emergency investigations of single geo-hazards can be divided into four steps, i.e., indoor preparation, site investigation, on-site fast processing and application, and indoor comprehensive processing and application. It should be noted that the automatic or manual scheme should be selected first during the on-site investigation, according to the environmental assessment. In addition, as long as there are more than five stable GNSS satellite signals in the geo-hazard area, the vast majority of emergency investigations should use the automatic scheme. The layout and measurement of GCPs is also vital in improving the accuracy of the photogrammetric processing results. The aim of fast processing is to support the rapid on-site development of preliminary emergency response plans for geo-hazards,

whereas the comprehensive processing, which is performed indoors, is intended to support the design of detailed emergency response plans. The SfM photogrammetric method is recommended for use regardless of whether the fast processing or the comprehensive processing is employed.

(3) Mastering the key techniques and methods contribute to better use of UAVs for emergency investigation of single geo-hazards. The following points are worth noting. Before the on-site flight and image collection, three to five GCPs should be established in or around the geo-hazard, and their distribution should be as uniform as possible, i.e., they should constitute equilateral triangles or a quadrilateral network. The RTK-DGPS technique should be used preferentially as long as there are stable GNSS signals, whereas total station measurement techniques would be a good choice in areas with unstable or even no GNSS signals. Proper route planning is key to ensure the safety and efficiency of UAV-based emergency investigations. Three typical route types are recommended. The planar grid pattern is suitable for geo-hazards that cover large areas on gentle slopes, the vertical grid pattern is suitable for geo-hazards that are developed on steep slopes, and the combined grid pattern is suitable for geo-hazards that occur on transitional terrain, which includes both gentle and steep slopes. In particular applications, the planned route should be selected, combined or flexibly changed from the three typical route types mentioned above, based on the spatial characteristics of individual geo-hazards. However, in any case, the planned route must ensure that the frontal overlap ratios of the obtained pictures are at least 75%, and the side overlap ratios should be at least 60%. It is essential to carry out a pre-flight inspection after importing the accurate planned route data and setting the flight parameters. Moreover, during flights, it is best to have three technical staff members on hand to ensure flight safety and image quality. When the SfM photogrammetric processing method is used, the results should also be targeted according to the types or characteristics of different geo-hazards.

The successful examples described in this paper demonstrate that using UAVs for emergency investigation of single geo-hazards can greatly reduce the time, intensity and risks associated with on-site work and provide valuable high-accuracy and high-resolution information that supports the development of emergency responses.

**Acknowledgements** This work was supported by the Hubei Provincial Natural Science Foundation of China (2017CFB436), the Open Research Fund of the Hubei Key Laboratory of Intelligent Vision Based Monitoring for Hydroelectric Engineering (2016KLA02), the Open Research Fund of the Key Laboratory of Disaster Prevention and Mitigation of Hubei Province (2016KJZ16), the Hubei Science and Technology Support Programme (2015BCE070, 2015BCE038), and the Innovation Groups Project of the Natural Science Foundation of Hubei Province (2015CFA025).

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

**List of Figures**

**Fig. 1.** A photograph showing the customized UAV system used for emergency investigation of single geo-hazards in the Three Gorges Reservoir area, China.

**Fig.2.** A diagram showing the architecture and main function modules of the customized UAV system.

**Fig.3.** The process of using UAVs to emergency investigate single geo-hazards.


**Fig.4.** The planning of a route involving a typical regular grid pattern in plane view using Mission Planner software.

**Fig.5.** Three typical route types for UAV-based emergency investigation of single geo-hazards. (a) Planar grid pattern for slightly inclined slopes; (b) vertical grid pattern for steep slopes; (c) combined grid pattern for transitional terrain.


**Fig.6.** The results of a UAV-based emergency investigation of a slightly inclined landslide. (a) the digital orthophoto; (b) the 3D texture model; (c) interpretation of the destruction.

**Fig.7.** The results of a UAV-based emergency investigation for a dangerous rock mass on steep cliff. (a) Overview photo; (b) right side 685 photo of the dangerous rock mass; (c) 3D texture model of the whole scene; (d) 3D texture model of the dangerous rock mass.

**Fig.8.** The results of a UAV-based emergency investigation for a combined slope on transitional terrain. (a) The 3D texture model and the identified potential geo-hazards; (b) the quantitative characteristics of the steep slope above the highway.




**List of Tables**

**Table 1.** The core components and main features of the customized UAV system.

       **Table 2.** The time required by each step of the workflow in the three sample applications.







**Figures**


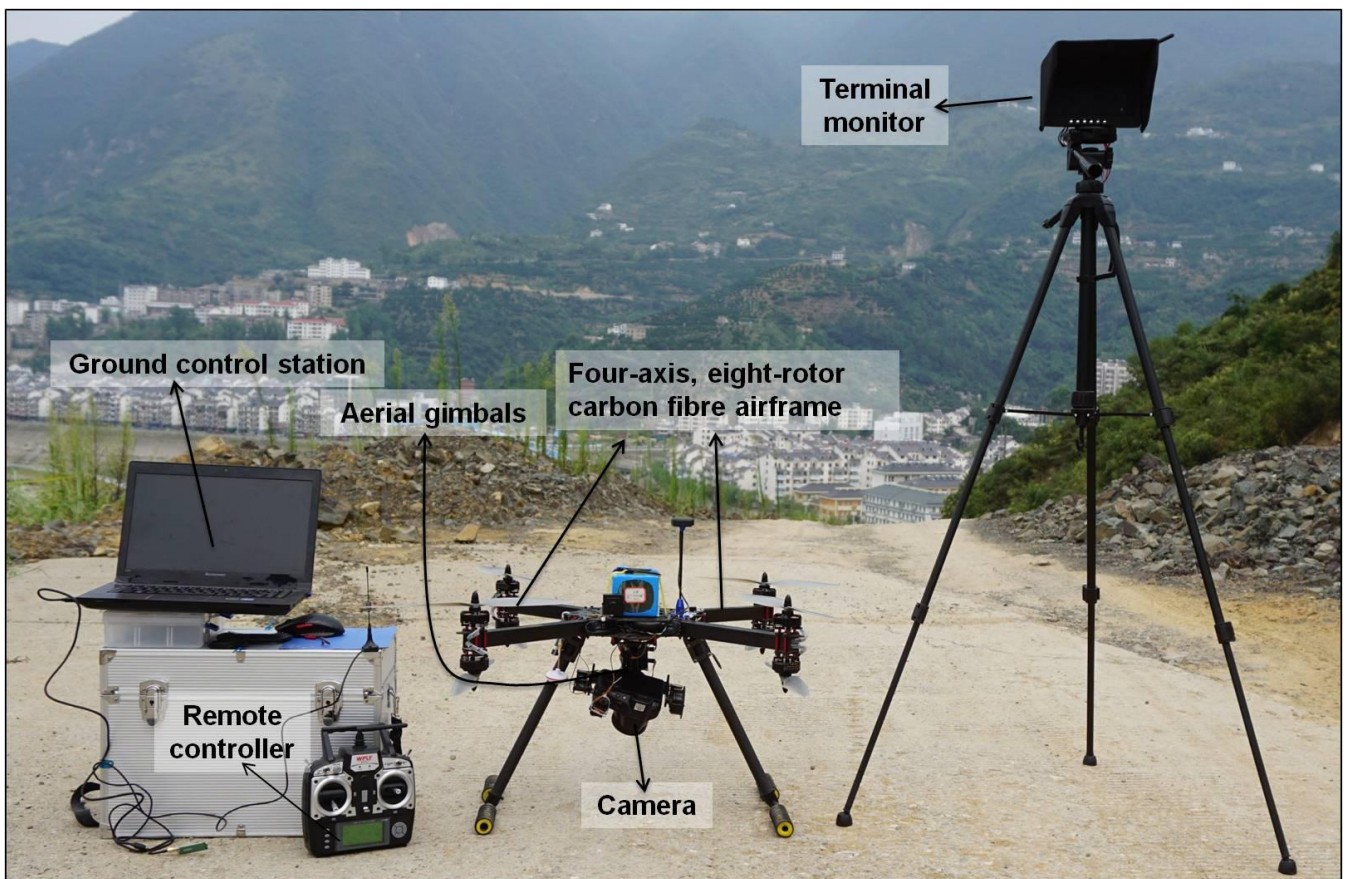

**Fig.1.** A photograph showing the customized UAV system used for emergency investigation of single geo-hazards in the Three Gorges Reservoir area, China.


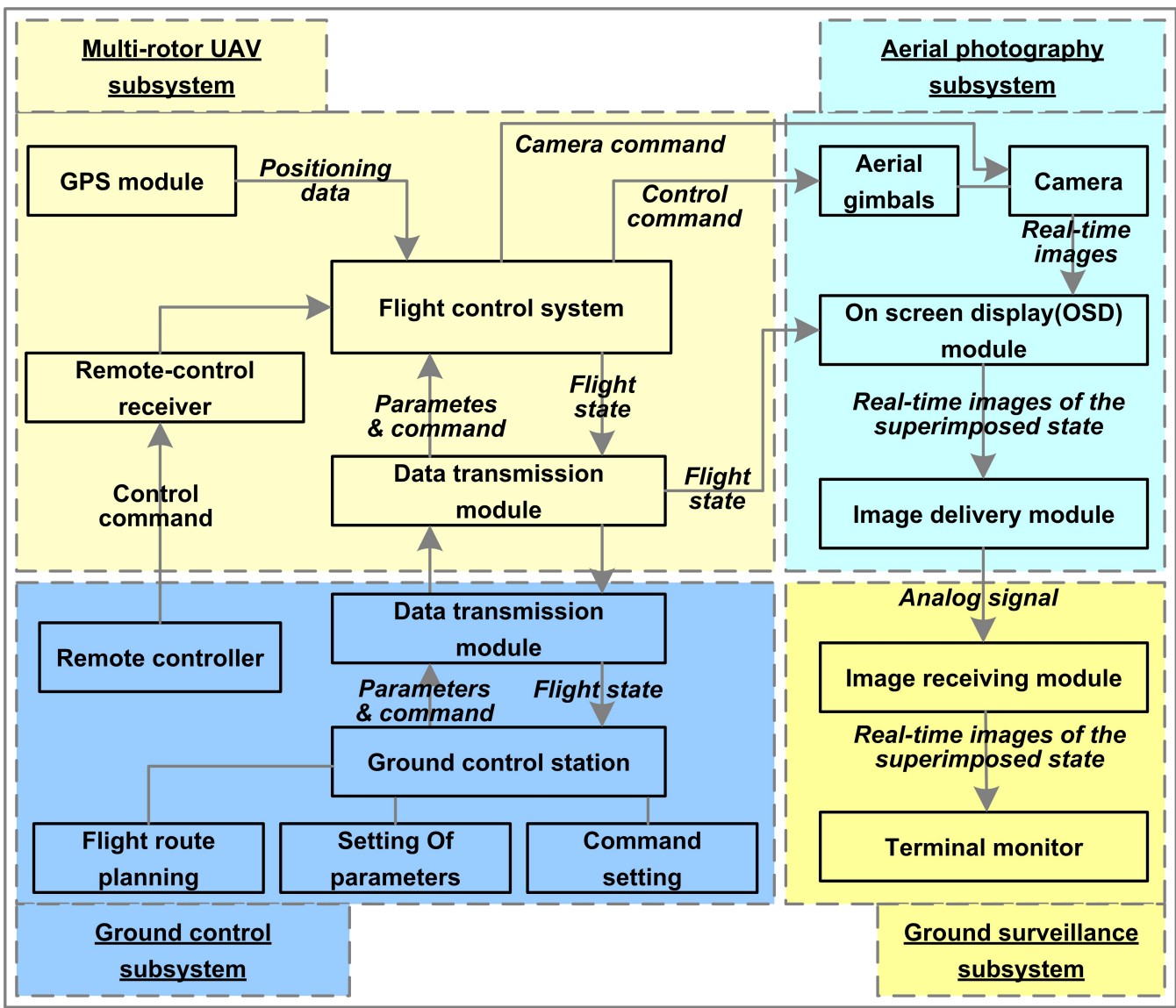


**Fig.2.** A diagram showing the architecture and main function modules of the customized UAV system.

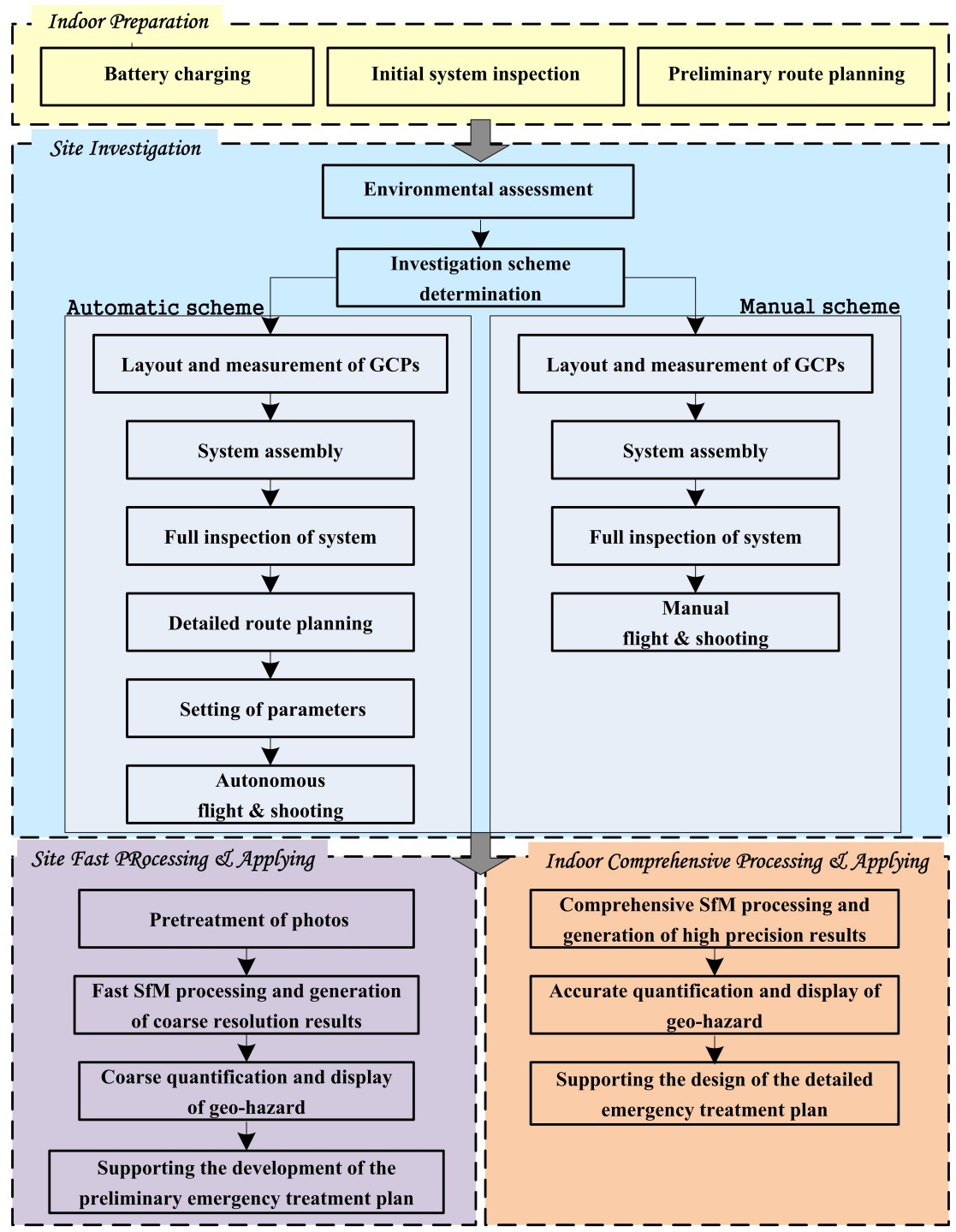

**Fig.3.** The process of using UAVs to emergency investigate single geo-hazards.

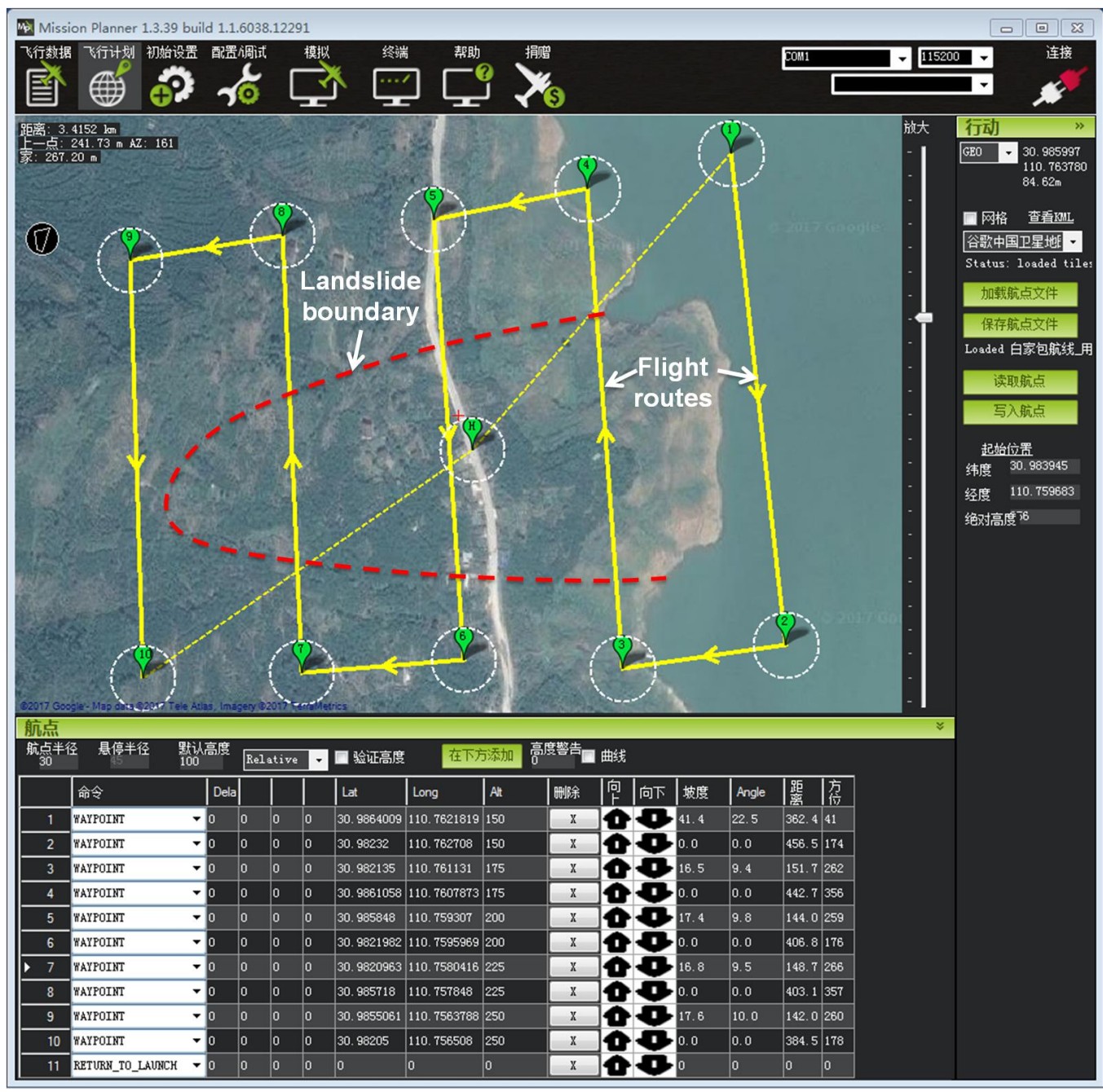

**Fig.4.** The planning of a route involving a typical regular grid pattern in plane view using Mission Planner software.

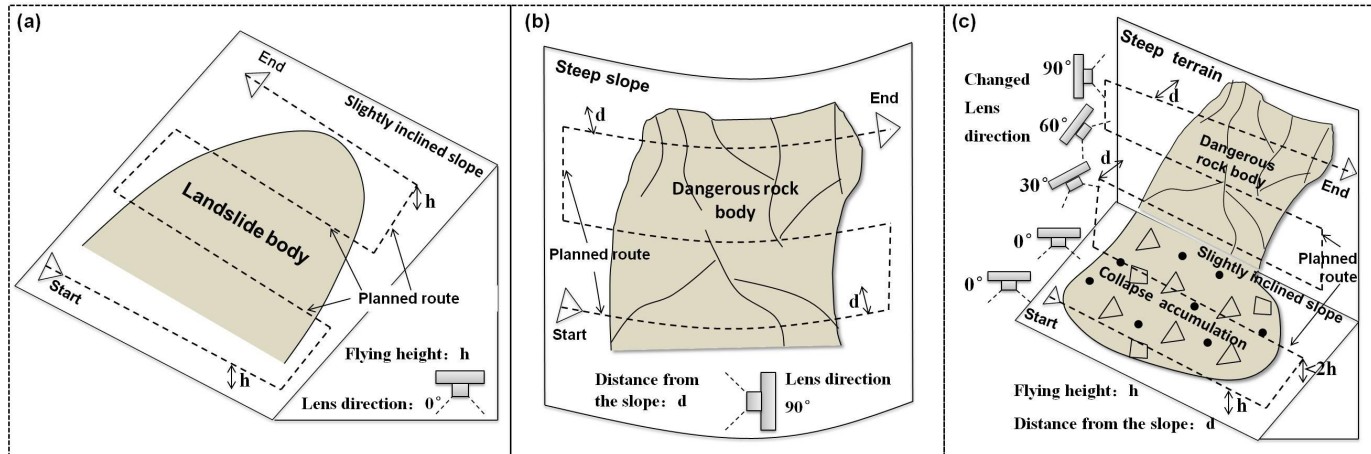

**Fig.5.** Three typical route types for UAV-based emergency investigation of single geo-hazards. (a) Planar grid pattern for slightly inclined slopes; (b) vertical grid pattern for steep slopes; (c) combined grid pattern for transitional terrain.

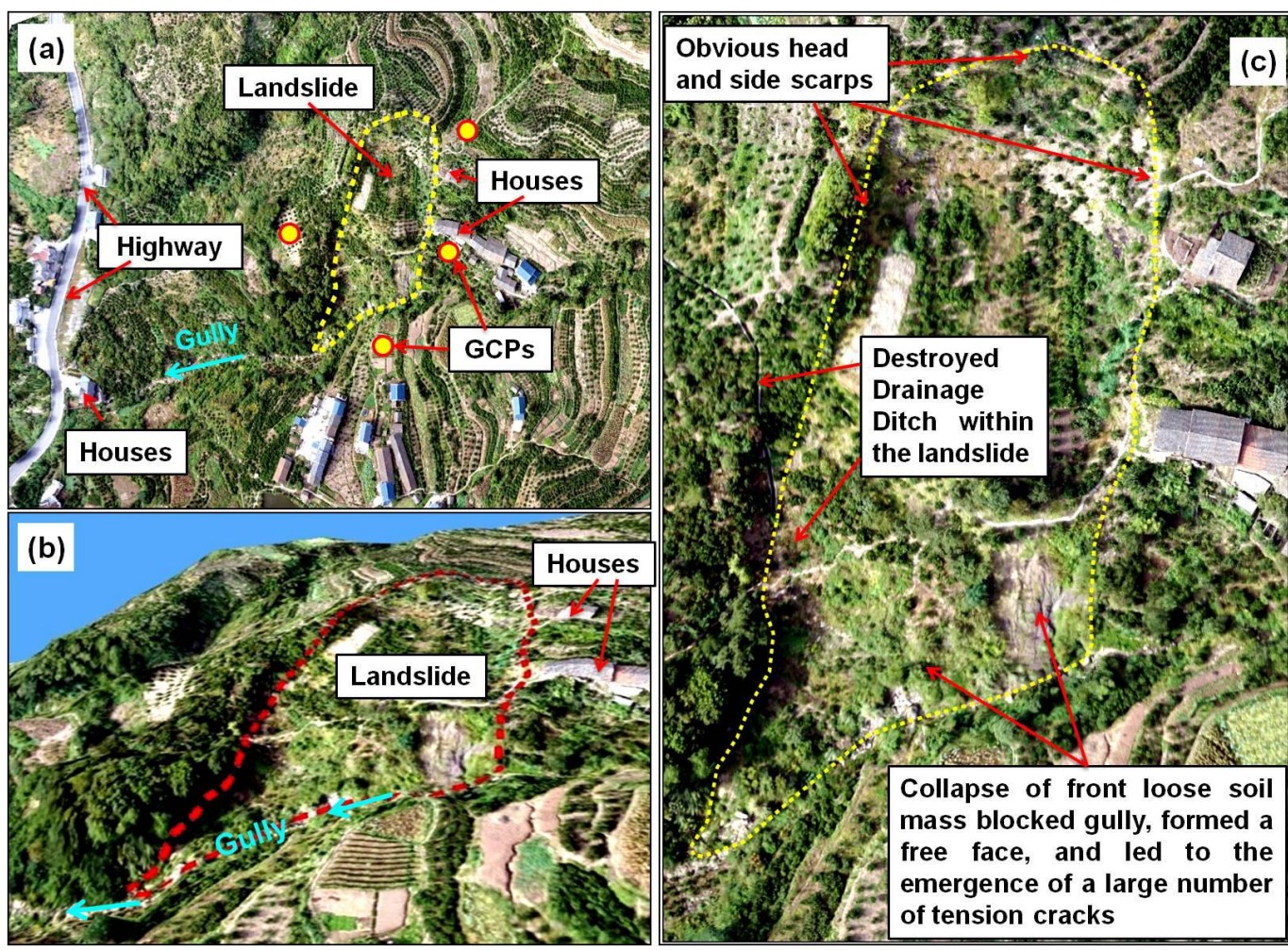

**Fig.6.** The results of a UAV-based emergency investigation of a slightly inclined landslide. (a) the digital orthophoto; (b) the 3D texture model; (c) interpretation of the destruction.

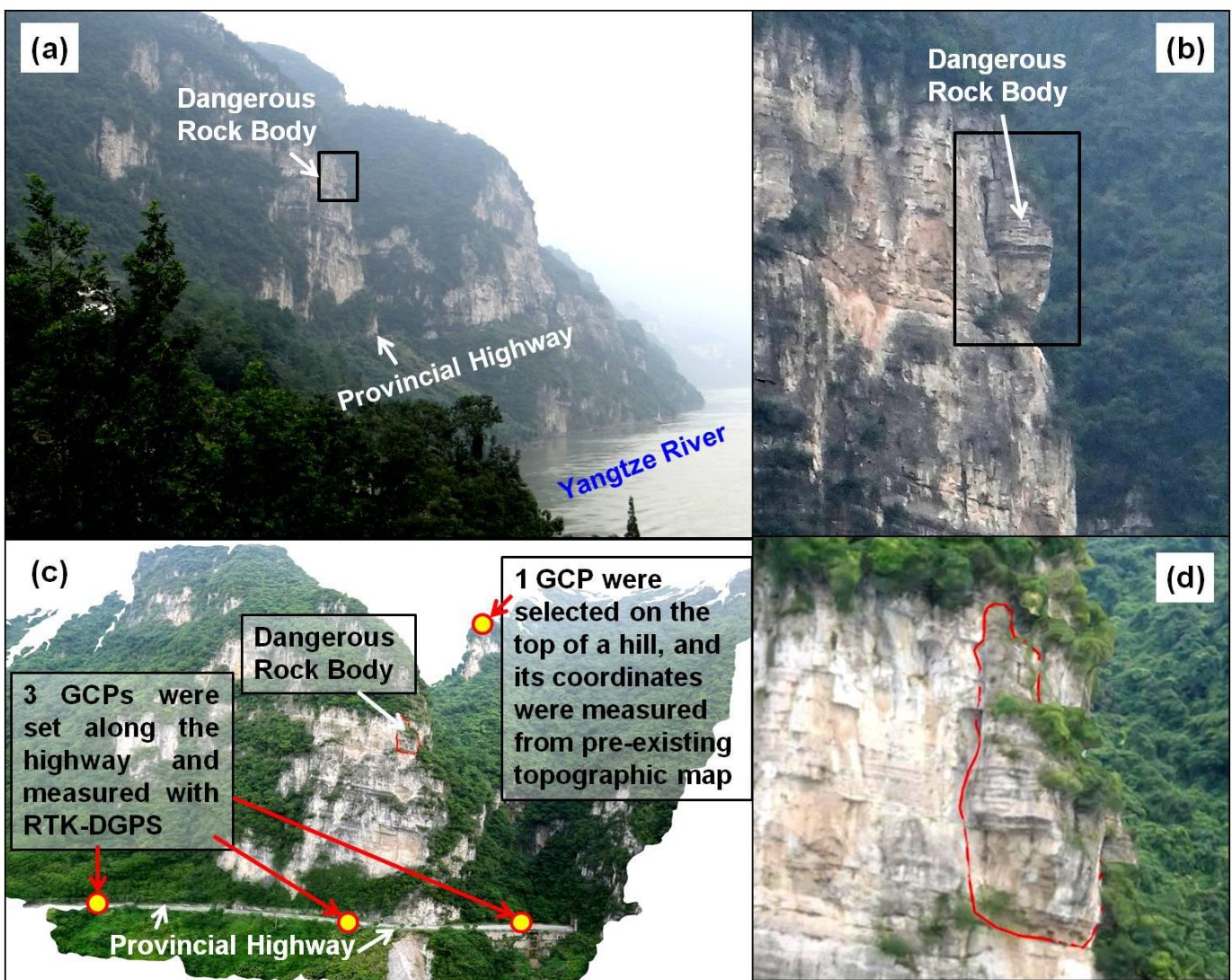

**Fig.7.** The results of a UAV-based emergency investigation for a dangerous rock mass on steep cliff. (a) Overview photo; (b) right side photo of the dangerous rock mass; (c) 3D texture model of the whole scene; (d) 3D texture model of the dangerous rock mass.

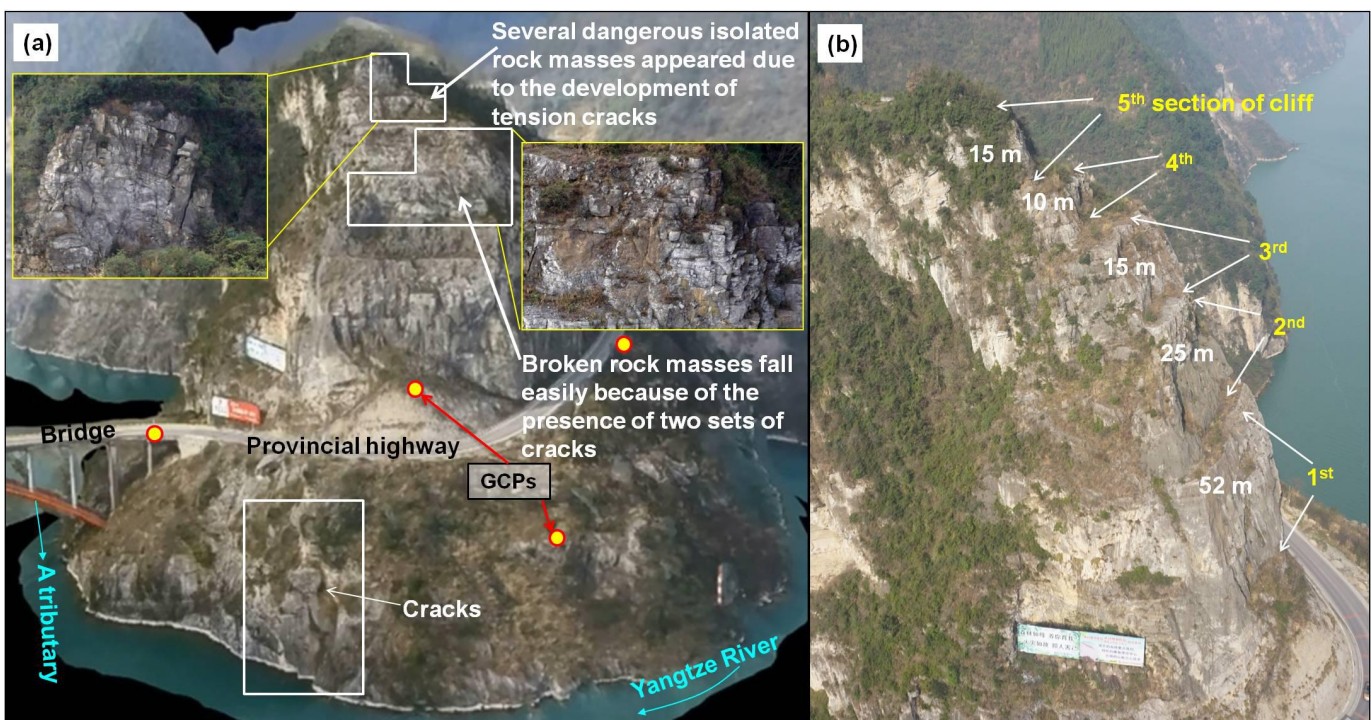

**Fig.8.** The results of a UAV-based emergency investigation for a combined slope on transitional terrain. (a) The 3D texture model and the identified potential geo-hazards; (b) the quantitative characteristics of the steep slope above the highway.




 **Tables**

**Table 1.** The core components and main features of the customized UAV system.

| Subsystem | Core component | Functions | Parameters | Features |
|---|---|---|---|---|
| Multi-rotor UAV subsystem | UAV airframe | Flying and carrying the various components, e.g., aerial photography subsystem | Customized four-axis & eight-rotors carbon fiber airframe | High strength, light weight; quick folding and convenient assembling & disassembling ; small size, large carrying capacity; good maneuverability and high hover efficiency. |
| | Flight control system | The brain of the UAV system that gives support to the flight safety, and ensure that the UAV is stable and can be manipulated in all flight phases and conditions. | Open source flight control system, i.e., Pixhawk 2.4.5 (Meier et al., 2012), with dual processors, a 32-bit core processor that is used for computing, and another coprocessor for failure protection even if the main one crashed. | Small size, light weight; low power consumption, high integration; high robustness, powerful data processing and real-time communication; open source with good extensibility; perfect matched, open source and free ground control software. |
| Aerial photography subsystem | Camera | Taking high-resolution photos. | Sony HX200 digital camera: Vario Sonnar T* 4.8-144mm F/2.8–5.6 lens; 18.2 million effective pixels. | High resolution, supporting a variety of shooting modes. |
| | Aerial gimbals | Keep camera stability to ensure clear shooting, and accurately adjust the lens orientation according to actual needs. | 3 axis brushless gimbal ptz | Small size, light weight, high precision and good versatility. |
| Ground control subsystem | Ground control station | Interacting with flight control system, to achieve the UAV's debugging and maintenance, parameter settings, route planning, monitoring and control, etc. | Open source ground station software, i.e., Mission Planner 1.3.39 (Team, 2016). | Match with flight control system PIXHAWK, free open source, support for Windows operation system. |
| | Remote controller | Manual control of UAV flight and camera shooting. | 2.4GHz Seven-channel full-scale remote controller | Reliable quality, cost-effective. |
| Ground surveillance subsystem | Image transmission | Real-time transmission of UAV aerial images with flight state data to the terminal monitor. | FPV image transmission system (TS832/RC832), with 5.8G, 32channels, 600MW. | Small size, stable signal, far transmission distance |
| | Terminal monitor | Real-time receiving and displaying UAV flight image and all flight state data. | 7-inch monitor display | Ultra-thin, high integration |

**Table 2.** The time required by each step of the workflow in the three sample applications. (Unit: minutes)

| Main step | A slightly inclined landslide | A dangerous rock mass on a steep cliff | A combined slope on transitional terrain |
|---|---|---|---|
| UAV preparation,flight and photo image collection | 30 | 35 | 50 |
| Layout and measurement of GCPs | 50 | 25 | 40 |
| **UAV-based investigation** | **50** | **35** | **50** |
| On-site fast processing | 28 | / | 40 |
| **On-site work** | **78** | **35** | **90** |
| Indoor comprehensive processing | 65 | 100 | 95 |
| **Total** | **143** | **135** | **185** |
