# Peer review of "A method for using unmanned aerial vehicles for emergency investigation of single geo-hazards and sample applications of this method"

_Natural Hazards and Earth System Sciences, 2017_

## Referee Comment (RC1) · Anonymous Referee #1 · 4 Mar 2017

In general

The paper describes a drone specifically design by the authors for emergency investigations. This UAV was been applied in 3 practical cases to demonstrate its efficacy. In my opinion, there are 3 limits in this paper:  c The paper contains no mentions of Direct Photogrammetry (DF) approach, but In a drone specifically designed for emergencies and rapid mapping, this approach has to be applied. I suggest the use of DP techniques to measure directly in field external orientation parameters and the application of a post processing BBA to refine the external orientation parameters directly measured. Could this UAV be equipped with sensors for DF? In the case of affirmative answer, I suggest to the authors to include some

details of this solution (kind of IMU/GNSS sensors, real time or post processing, used software tools, and so on); • Paper don't describe innovative approach to SfM survey using UAV: merely, there are some details of practical suggestions for UAV survey and some reports of applicative examples, actually known in scientific literature. To complete these descriptions, I suggest to complete the practical details including, in Paragraphs 5.1, 5.2 and 5.3, some information on number of acquired image, flight plan, time spent for the acquisition and post processing, number of points of dense point clouds, density of point cloud, obtained accuracy, ….; • The references don't include some important papers on the use of UAV for mapping, environmental application, rapid mapping and emergency investigation. For examples, I suggest: http://link.springer.com/article/10.1007/s12518-014-0144-x http://www.mdpi.com/1424-8220/15/7/15717 http://www.mdpi.com/2072-4292/8/9/779 http://www.tandfonline.com/doi/pdf/10.1080/19475705.2016.1225229

Detailed corrections: - Rows 110, 172, 179, 183, 220, 223, 246, 290, 301, 381, 384, 396, 411, 433, 467 Replace GPS with GNSS - Row 320 Replace facade in façade

---

## Referee Comment (RC2) · Anonymous Referee #2 · 7 Mar 2017

General advice:

This paper aims to describe a RPAS and processing pipeline specifically developed for the management of small hazard events. Authors discuss both the platform/sensor technology and the main steps followed during the complete UAV mission workflow. Finally, performance evaluation is carried out on three test cases. Although the core concept is interesting and may represent an interesting issue for the scientific community, several main issues should be addressed by the authors.

Major Comments:

1.General remark: the English is very poor and this may prevent a full comprehension

of the paper. Photogrammetry-related terminology is vague and often incorrect (e.g. "high-definition photos", "...for the photos, the definition, scope and overlap rate...", "planar digital terrain", etc...). A proofreading by a native English speaker conversant with photogrammetric terminology is strongly required.

2.The scientific significance and novelty of the paper should be proved. Which are the advantages of the developed platform/sensor/pipeline compared to other commercial or in-house developed systems? The literature review addresses only general concepts and does not show the novelty and advantages of the newly developed system.

3.The application field is vague. Authors say that the RPAS is developed for emergency investigation of "single" geo-hazards. What do you mean with the term "single"? If it refers to a limited spatial extension of the natural hazard, this should be better clarify and a clear idea of the intended area size should be given.

4.No accuracy figures are given. Authors generally refer to "meter-level error" or "centimeter- even millimeter- level accuracy". How did you evaluate accuracy? Did you adopt Control Points to check the accuracy of orientation results? Did you evaluate the accuracy of the final product? Although accuracy is not the main aim of rapid mapping, a metric evaluation of the methodology is necessary to confirm and support the conclusions.

5.Why is direct geo-referencing not dealt with?

6.The experimental part is very poor. No details are given regarding the image dataset (GSD?), the accuracy achieved, the time required. This gives limited support to the conclusion drawn by the authors.

---

## Editor Comment (EC1) · F. Nex (Editor) · 3 Apr 2017

Dear Authors, considering the comments received from the reviewers, I ask you to revise the manuscript according to their comments. Please upload a revised version of the paper, with detailed answers to the reviewers' comments. Please try to complete the work in three weeks time.

---

## Author Comment (AC1) · 8 May 2017

Authors: Thank you for consideration of our paper. The respond to your comment is as follows:

GENERAL COMMENTS Dear Authors, considering the comments received from the reviewers, I ask you to revise the manuscript according to their comments. Please upload a revised version of the paper, with detailed answers to the reviewers' comments. Please try to complete the work in three weeks time.

Authors: Now, we are revising our manuscript according to the comments received from the reviewers. Because major revise is needed, and the language needs to be

improved and polished, so please give more time, thanks a lot.

---

## Author Comment (AC2) · 8 May 2017

Authors: Thank you for your interests about our paper and valuable comments to improve it. The responds to your comments are as follows:

GENERAL COMMENTS This paper aims to describe a RPAS and processing pipeline specifically developed for the management of small hazard events. Authors discuss both the platform/sensor technology and the main steps followed during the complete UAV mission workflow. Finally, performance evaluation is carried out on three test cases. Although the core concept is interesting and may represent an interesting issue for the scientific community, several main issues should be addressed by the authors.

[Figure]

SPECIFIC COMMENTS The English is very poor and this may prevent a full compre-
hension of the paper. Photogrammetry-related terminology is vague and often incorrect
(e.g. "high-definition photos", "...for the photos, the definition, scope and overlap rate...",
"planar digital terrain", etc...). A proofreading by a native English speaker conversant
with photogrammetric terminology is strongly required.

Authors: Thank you for the comment. After modifying the contents of the paper, we will
invite a native English speaker conversant with photogrammetric terminology, to help
us improve the English writing of the revised paper.

The scientific significance and novelty of the paper should be proved. Which are the ad-
vantages of the developed platform/sensor/pipeline compared to other commercial or
in-house developed systems? The literature review addresses only general concepts
and does not show the novelty and advantages of the newly developed system.

Authors: Thank you for the comment about the scientific significance and novelty of
the paper. In fact, The main aim of this paper is to conclude and establish a complete
method of using UAV for emergency investigation of small hazard events. In the revised
paper, we will strengthen the literature review about this aspect.

The application field is vague. Authors say that the RPAS is developed for emergency
investigation of "single" geo-hazards. What do you mean with the term "single"? If it
refers to a limited spatial extension of the natural hazard, this should be better clarify
and a clear idea of the intended area size should be given.

Authors: Thank you very much for the comment and suggestion. Indeed, the "single"
geo-hazard refers to a limited spatial extension of a natural hazard, so we will give a
better clarify and a clear idea of the intended area size in the revised paper.

No accuracy figures are given. Authors generally refer to "meter-level error" or
"centimeter- even millimeter- level accuracy". How did you evaluate accuracy? Did
you adopt Control Points to check the accuracy of orientation results? Did you evaluate the accuracy of the final product? Although accuracy is not the main aim of rapid mapping, a metric evaluation of the methodology is necessary to confirm and support the conclusions. Authors: Thank you for the comment about the accuracy. And the accuracy is indeed an important indicator of the availability of results, in our method, the GCPs were usually used for accuracy assessment, simply, the root-mean-square error (RMSE) of GCPs was often used as an important indicator. So, we will add the accuracy results in 5. three application examples.

Why is direct geo-referencing not dealt with?

Authors: In fact, the direct geo-referencing is used in our method, especially in the site investigation and the site fast processing. Specifically, when the GNSS signal can be used during the site investigation, the location information will be automatically wrote into the captured photos, to ensure that the use of fast SfM processing method can generate geo-referencing results. If there is no GNSS signal, the GCPs layout and measurement is indispensable to support the SfM photogrammetric processing, i.e., introducing GCPs to ensure generate geo-referencing results. So, some detailed processing method, such as SfM and so on will be added to the revised paper.

The experimental part is very poor. No details are given regarding the image dataset (GSD?), the accuracy achieved, the time required. This gives limited support to the conclusion drawn by the authors.

Authors: Thank you for the comment. More practical details including the number of acquired image, time spent for the acquisition and post processing, number of points of dense point clouds, density of point cloud, obtained GSD and accuracy, etc., will be added in the revised paper.

---

## Author Comment (AC3) · 8 May 2017

Authors: Thank you for your interests about our paper and valuable comments to improve it. The responds to your comments are as follows:

GENERAL COMMENTS The paper describes a drone specifically design by the authors for emergency investigations. This UAV was been applied in 3 practical cases to demonstrate its efficacy. In my opinion, there are 3 limits in this paper:

SPECIFIC COMMENTS The paper contains no mentions of Direct Photogrammetry (DF) approach, but in a drone specifically designed for emergencies and rapid mapping, this approach has to be applied. I suggest the use of DP techniques to measure

directly in field external orientation parameters and the application of a post processing BBA to refine the external orientation parameters directly measured. Could this UAV be equipped with sensors for DF? In the case of affirmative answer, I suggest to the authors to include some details of this solution (kind of IMU/GNSS sensors, real time or post processing, used software tools, and so on);

Authors: Thank you for commenting and suggesting about the DF. Actually, although the DF is not mentioned in the paper, this approach is used, especially in the site investigation and the site fast processing. Specifically, when the GNSS signal can be used during the site investigation, the location information will be automatically wrote into the captured photos, to ensure that the use of fast SfM processing method can generate coarse-precision results with a real space coordinate system in the site fast processing step. If there is no GNSS signal, the GCPs layout and measurement is indispensable to support the SfM photogrammetric processing, i.e., introducing GCPs to ensure generate results with a real space coordinate system. So, according to the suggestion, the details of DP, including IMU/GNSS sensors, post processing, used software tools, and so on will be added to the revised paper.

Paper don't describe innovative approach to SfM survey using UAV: merely, there are some details of practical suggestions for UAV survey and some reports of applicative examples, actually known in scientific literature. To complete these descriptions, I suggest to complete the practical details including, in Paragraphs 5.1, 5.2 and 5.3, some information on number of acquired image, flight plan, time spent for the acquisition and post processing, number of points of dense point clouds, density of point cloud, obtained accuracy.

Authors: Thank you for the suggestion. These detailed information will be added in the revised paper.

The references don't include some important papers on the use of UAV for mapping, environmental application, rapid mapping and emergency investigation.

For examples, I suggest: http://link.springer.com/article/10.1007/s12518-014-0144-x
http://www.mdpi.com/1424-8220/15/7/15717 http://www.mdpi.com/2072-4292/8/9/779
http://www.tandfonline.com/doi/pdf/10.1080/19475705.2016.1225229

Authors: Thank you for the suggestion. These references will be carefully read and
added to the appropriate location in the revised paper.

Detailed corrections: - Rows 110, 172, 179, 183, 220, 223, 246, 290, 301, 381, 384,
396, 411, 433, 467 Replace GPS with GNSS - Row 320 Replace facade in façade

Authors: Thank you for the corrections. These comments will be reflected in the revised
paper.

---

## Author Response (AR1)

Response to EC1 again:

Authors: Thank you for consideration of our paper. The respond to your comment is as flowing:

Dear Authors, considering the comments received from the reviewers, I ask you to revise the manuscript according to their comments. Please upload a revised version of the paper, with detailed answers to the reviewers' comments. Please try to complete the work in three weeks time.

**Authors:** Now, we have revised our manuscript according to the comments received from the reviewers, at the same time, a point-by-point reply to the comments and a marked-up manuscript version showing the changes had been uploaded.

Besides point-by-point revise according to the comments of reviewers, another main changes include as follows:
1. We **changed the title** "**Method and application of using unmanned aerial vehicle for emergency investigation of single geo-hazard**" to "**A method for using unmanned aerial vehicles for emergency investigation of single geo-hazards and sample applications of this method**", may be more suitable.
2. We **invited native English speaker conversant with photogrammetric terminology, to help us improve the English writing of the revised paper.**
3. We **checked, modified, and improved all figures and tables.**

We would like to express our great appreciation to you and reviewers for comments on our paper. Looking forward to hearing from you.

Response to RC1 again:

Authors: Thank you for your interests about our paper and valuable comments to improve it. The responds to your comments are as flowing:

GENERAL COMMENTS The paper describes a drone specifically design by the authors for emergency investigations. This UAV was been applied in 3 practical cases to demonstrate its efficacy. In my opinion, there are 3 limits in this paper:

1. The paper contains no mentions of Direct Photogrammetry (DF) approach, but in a drone specifically designed for emergencies and rapid mapping, this approach has to be applied. I suggest the use of DP techniques to measure directly in field external orientation parameters and the application of a post processing BBA to refine the external orientation parameters directly measured. Could this UAV be equipped with sensors for DF? In the case of affirmative answer, I suggest to the authors to include some details of this solution (kind of IMU/GNSS sensors, real time or post processing, used software tools,

and so on);

Authors: Thank you for commenting and suggesting about the DF. Actually, although the DF is not mentioned in the paper, this approach is used, especially in the site investigation and the site fast processing. Specifically, when the GNSS signal can be used during the site investigation, the location information will be automatically wrote into the captured photos, to ensure that the use of fast SfM processing method can generate coarse-precision results with a real space coordinate system in the site fast processing step, please see lines 269-282. If there is no GNSS signal, the layout and measurement of GCPs is indispensable to support the SfM photogrammetric processing, i.e., introducing GCPs to ensure generate results with a real space coordinate system, please see lines 243-246, 299-308. According to the suggestion, the details of DP, including IMU/GNSS sensors (lines 270-282), post processing (lines 299-308), used software tools (lines 277-280, 304-306, 407-409) had been added to the revised paper.

2. Paper don't describe innovative approach to SfM survey using UAV: merely, there are some details of practical suggestions for UAV survey and some reports of applicative examples, actually known in scientific literature. To complete these descriptions, I suggest to complete the practical details including, in Paragraphs 5.1, 5.2 and 5.3, some information on number of acquired image, flight plan, time spent for the acquisition and post processing, number of points of dense point clouds, density of point cloud, obtained accuracy.

Authors: Thank you for the suggestion. These detailed information had been added in the revised paper, please see lines 424-438, 463-480, 497-506.

3. The references don't include some important papers on the use of UAV for mapping, environmental application, rapid mapping and emergency investigation. For examples, I suggest: http://link.springer.com/article/10.1007/s12518-014-0144-x
http://www.mdpi.com/1424-8220/15/7/15717 http://www.mdpi.com/2072-4292/8/9/779
http://www.tandfonline.com/doi/pdf/10.1080/19475705.2016.1225229

Authors: Thank you for the suggestion. These references had been carefully read and added to the appropriate location in the revised paper, please see lines 45-57.

Detailed corrections: - Rows 110, 172, 179, 183, 220, 223, 246, 290, 301, 381, 384, 396, 411, 433, 467 Replace GPS with GNSS - Row 320 Replace facade in façade

Authors: Thank you for the corrections. These comments had been reflected in the revised paper.

In addition, three main changes include as follows:
4.   We **changed the title** "**Method and application of using unmanned aerial vehicle for emergency investigation of single geo-hazard**" to "**A method for using unmanned aerial vehicles for emergency investigation of single geo-hazards and sample applications of this method**", may be more suitable.

5. We **invited native English speaker conversant with photogrammetric terminology, to help us improve the English writing of the revised paper.**
6. We **checked, modified, and improved all figures and tables.**

We have tried our best to revise our manuscript according to your valuable comments, and hope that the correction will meet with approval.

Response to RC2 again:

Authors: Thank you for your interests about our paper and valuable comments to improve it. The responds to your comments are as flowing:

GENERAL COMMENTS This paper aims to describe a RPAS and processing pipeline specifically developed for the management of small hazard events. Authors discuss both the platform/sensor technology and the main steps followed during the complete UAV mission workflow. Finally, performance evaluation is carried out on three test cases. Although the core concept is interesting and may represent an interesting issue for the scientific community, several main issues should be addressed by the authors.

1.General remark: the English is very poor and this may prevent a full comprehension of the paper. Photogrammetry-related terminology is vague and often incorrect (e.g. "high-definition photos", "...for the photos, the definition, scope and overlap rate...", "planar digital terrain", etc...). A proofreading by a native English speaker conversant with photogrammetric terminology is strongly required.

Authors: Thank you for the comment. After revised the contents of the paper, we had invited a native English speaker conversant with photogrammetric terminology, to help us improve the English writing of the revised paper.

2.The scientific significance and novelty of the paper should be proved. Which are the advantages of the developed platform/sensor/pipeline compared to other commercial or in-house developed systems? The literature review addresses only general concepts and does not show the novelty and advantages of the newly developed system.

Authors: Thank you for the comment about the scientific significance and novelty of the paper. In fact, The main aim of this paper is to conclude and establish a complete method of using UAV for emergency investigation of small hazard events. In the revised paper, we had strengthened the literature review about this aspect, please see lines 45-63, 70-79.

3.The application field is vague. Authors say that the RPAS is developed for emergency investigation of "single" geo-hazards. What do you mean with the term "single"? If it refers to a limited spatial extension of the natural hazard, this should be better clarify and a clear idea of the intended area size should be given.

Authors: Thank you very much for the comment and suggestion. Indeed, the "single" geo-hazard refers to a limited spatial extension of a natural hazard, so we add a better clarify and a clear idea of the intended area size in the revised paper, please see lines 61-63.

4.No accuracy figures are given. Authors generally refer to "meter-level error" or "centimeter- even millimeter- level accuracy". How did you evaluate accuracy? Did you adopt Control Points to check the accuracy of orientation results? Did you evaluate the accuracy of the final product? Although accuracy is not the main aim of rapid mapping, a metric evaluation of the methodology is necessary to confirm and support the conclusions. Authors: Thank you for the comment about the accuracy. And the accuracy is indeed an important indicator of the availability of results, in our method, the GCPs were used for accuracy assessment, simply, the root-mean-square error (RMSE) of GCPs was used as an important indicator. So, we add the accuracy results in 5. three application examples, please see lines 430-437, 472-477, 502-505.

5.Why is direct geo-referencing not dealt with?

Authors: In fact, the direct geo-referencing is used in our method, especially in the site investigation and the site fast processing. Specifically, when the GNSS signal can be used during the site investigation, the location information will be automatically wrote into the captured photos, to ensure that the use of fast SfM processing method can generate geo-referencing results. If there is no GNSS signal, the GCPs layout and measurement is indispensable to support the SfM photogrammetric processing, i.e., introducing GCPs to ensure generate geo-referencing results. Accordingly, above detailed processing method, such as SfM and so on had been added to the revised paper, please see lines 270-274, 277-282, 300-301, 304-308, 402-409.

6.The experimental part is very poor. No details are given regarding the image dataset (GSD?), the accuracy achieved, the time required. This gives limited support to the conclusion drawn by the authors.

Authors: Thank you for the comment. More practical details including the number of acquired image, time spent for the acquisition and post processing, obtained GSD and accuracy, etc., had been added in the revised paper, please see lines 424-438, 463-480, 497-506.

In addition, We **changed the title "Method and application of using unmanned aerial vehicle for emergency investigation of single geo-hazard" to "A method for using unmanned aerial vehicles for emergency investigation of single geo-hazards and sample applications of this method",** may be more suitable. Moreover, We **checked, modified, and improved all figures and tables.**

We have tried our best to revise our manuscript according to your valuable comments,

and hope that the correction will meet with approval.

**A method for using unmanned aerial vehicle for emergency investigation of single geo-hazard and sample applications of this method**

**Abstract.** In recent years,  unmanned aerial vehicle (UAV) become widely used in  emergency investigation of major natural hazards  over  large area; 
[revised manuscript text omitted]

---

## Referee Report (RR1)

Comments on the paper "*A method for using unmanned aerial vehicles for emergency investigation of single geo-hazards and sample applications of this method*" submitted to Natural Hazards and Earth System Sciences.

Manuscript ID: nhess-2017-44

Iteration: Major Revision

General advice:

This paper aims to describe a RPAS and processing pipeline specifically developed for the management of small hazard events. Authors discuss both the platform/sensor technology and the main steps followed during the complete UAV mission workflow. Finally, performance evaluation is carried out on three test cases. The revised version of the manuscript correctly addressed the major concerns highlighted during the first revision round. However, few minor issues should be further considered by the authors.

Comments:

1.  Please, replace "high-definition" with "high-resolution" throughout the text.

2.  Line 45: please, remove "of features".

3.  Line 122-123: please, replace "new digital photogrammetric technologies" with "tools for semi-automated photogrammetric processing". Furthermore, the authors are encouraged to include the following recent works among the references dealing with photogrammetric techniques coupled with computer vision approaches and, in particular, the SfM approach:

Förstner, W. and Wrobel, B., 2016. *Photogrammetric computer vision.* Springer.

Özyeşil, O., Voroninski, V., Basri, R. and Singer, A., 2017. *A survey of structure from motion.* Acta Numerica, 26, pp.305-364.

4.  Lines 145-146: the authors make a comparison between their own system and a popular commercial one. Although interesting, the remarks about the improvements in image quality and system performance should be adequately proven.

5.  Line 191: please, replace "scheme" with "schemes".

6.  Line 255: please, correct the typo "fast  processing fast processing"

7.  Line 270: please, replace "SfM" with "semi-automated SfM-based".

8.  Line 274: please, replace "The fast SfM" with "The adopted SfM-based".

9.  Line 275: it is not clear whether the entire photogrammetric processing is performed using images at reduced resolution or only the tie point extraction step is carried out on images at lower resolution. Please, clarify this issue. Furthermore, since GPS data are exploited in the BBA, please add "GPS-assisted" aerial triangulation.

10. Line 326: since the (minimum) number and distribution of GCPs are essential factors to be considered when balancing requirements in terms of accuracy and efficiency, authors are encouraged to add references that further support/discuss what is here stated.

11. Line 349: if the flight altitude increases, the mean GSD (ground sampling distance) of the acquired images is bigger, thus providing for lower spatial resolution of the final photogrammetric products. Please, improve your sentence.

12. Line 408: replace "2D DSMs" with "2.5D DSMs".

13. Section 5 - Application examples.
    a. Authors are encouraged to add a table with the time required by each step of the workflow in order to provide a quick overview and a means of comparison among the three different experimental tests.
    b. Please, add the mean GSD of the original images.
    c. Please, use the term "spatial resolution" when referring to the final photogrammetric products (orthophoto and DSM) and "GSD" when referring to the original images.
    d. It is not clear whether the accuracy assessment is performed on the orientation results of the aerial triangulation or on the final orthophoto. Please, comment on this. Furthermore, please remark that, although a rigorous accuracy assessment should be performed by using external and independent check points, the GCPs themselves are here exploited given the limited availability of external ground truth. Finally, please report on the accuracy of the GCPs' 3D coordinates measured in the field.

---

## Author Response (AR2)

*Response to EC:*

Authors: Thanks very much for your kind work and consideration on publication of our paper. The respond to your comment is as flowing:

Dear Authors,
The paper has been improved, as stated in the reviewer's comments. Please, modify the paper according to the reviewer's correction before sending the new version.

Authors: Dear Editor, we have revised our manuscript according to the reviewer's comments, and with a point-by-point reply to the comments, a marked-up manuscript version showing the changes made.

*Response to nhess-2017-44-referee-report-1:*

Authors: Thank you very much for your interests about our paper and valuable comments to improve it. The responds to your comments are as flowing:

General advice: This paper aims to describe a RPAS and processing pipeline specifically developed for the management of small hazard events. Authors discuss both the platform/sensor technology and the main steps followed during the complete UAV mission workflow. Finally, performance evaluation is carried out on three test cases. The revised version of the manuscript correctly addressed the major concerns highlighted during the first revision round. However, few minor issues should be further considered by the authors.

Comments:

1. Please, replace "high-definition" with "high-resolution" throughout the text.

Authors: This correction had been reflected in lines 27, 121, 306, 313, 319, 521, 530, 580 and in Table 1.

2. Line 45: please, remove "of features".

Authors: We have removed line 45.

3. Line 122-123: please, replace "new digital photogrammetric technologies" with "tools for semi-automated photogrammetric processing". Furthermore, the authors are encouraged to include the following recent works among the references dealing with photogrammetric techniques coupled with computer vision approaches and, in particular, the SfM approach:

Förstner, W. and Wrobel, B., 2016. Photogrammetric computer vision. Springer.

Özyeşil, O., Voroninski, V., Basri, R. and Singer, A., 2017. A survey of structure from motion. Acta Numerica, 26, pp.305-364.

Authors: The change had been reflected in lines 122-123. And the references had been carefully read and added to the appropriate location in line 124.

4. Lines 145-146: the authors make a comparison between their own system and a popular commercial one. Although interesting, the remarks about the improvements in image quality and system performance should be adequately proven.

Authors: We have added the remarks about the improvements in image quality ("the UAVs are usually used to take wide-angle photos, so with the same aperture (f 2.8), the shorter 4.8 mm focal length lens of Sony HX200 can capture more clear photos than the DJI Phantom 4 pro with a 8.8 mm focal length lens, because the latter has a more serious background blur.") and system performance ("due the stronger power and greater weight, the customized UAV displays more flexible control and better wind resistance.") in the revised paper, please see lines 145-148.

5. Line 191: please, replace "scheme" with "schemes".

Authors: This correction had been reflected in line 193.

6. Line 255: please, correct the typo "fast processing fast processing".

Authors: This correction had been reflected in line 257.

7. Line 270: please, replace "SfM" with "semi-automated SfM-based".

Authors: This correction had been reflected in line 272.

8. Line 274: please, replace "The fast SfM" with "The adopted SfM-based".

Authors: This correction had been reflected in line 276.

9. Line 275: it is not clear whether the entire photogrammetric processing is performed using images at reduced resolution or only the tie point extraction step is carried out on images at lower resolution. Please, clarify this issue. Furthermore, since GPS data are exploited in the BBA, please add "GPS-assisted" aerial triangulation.

Authors: Thank you for the suggestions. The "extracting tie points from the photographs at lower resolution" was clarified in line 277, and the "GPS-assisted" was added in line 278.

10. Line 326: since the (minimum) number and distribution of GCPs are essential factors to be

considered when balancing requirements in terms of accuracy and efficiency, authors are encouraged to add references that further support/discuss what is here stated.

Authors: Thank you for the suggestion. We added a remark "in order to balance requirements in terms of accuracy and efficiency, according to our practical experience, …" (lines 329-330), and added two supported references (lines 331-332).

11. Line 349: if the flight altitude increases, the mean GSD (ground sampling distance) of the acquired images is bigger, thus providing for lower spatial resolution of the final photogrammetric products. Please, improve your sentence.

Authors: Thank you for the suggestion. We changed the sentence to "conversely, higher flights increase the mean ground sampling distance (GSD) of the acquired images, thus providing for lower spatial resolution of the final photogrammetric products" (lines 353-354), to be more accurately expressed the meaning.

12. Line 408: replace "2D DSMs" with "2.5D DSMs".

Authors: This correction had been reflected in line 412.

13. Section 5 - Application examples.
    a. Authors are encouraged to add a table with the time required by each step of the workflow in order to provide a quick overview and a means of comparison among the three different experimental tests.

    Authors: Thank you for the suggestion. We added Table 2 in line 810 and was referenced in line 533.

    b. Please, add the mean GSD of the original images.

    Authors: We added this data in lines 432, 478-479, 511.

    c. Please, use the term "spatial resolution" when referring to the final photogrammetric products (orthophoto and DSM) and "GSD" when referring to the original images.

    Authors: This correction had been reflected in lines 434, 482, 513.

    d. It is not clear whether the accuracy assessment is performed on the orientation results of the aerial triangulation or on the final orthophoto. Please, comment on this. Furthermore, please remark that, although a rigorous accuracy assessment should be performed by using external and independent check points, the GCPs themselves are here exploited given the limited availability of external ground truth. Finally, please report on the accuracy of the GCPs' 3D coordinates measured in the field.

Authors: In fact, the accuracy assessment was performed on the final orthophoto by calculating the mean, standard deviation (*Sigma*), and root-mean square error (*RMSE*) values of all GCPs, so we added these detailed results to the revised paper, please see lines 436~440, 441~444, 483~485, 515~519. At the same time, we explained clearly: "Usually, a rigorous accuracy assessment should be performed by using external and independent check points, but for simplicity and efficiency, the 4 GCPs were used also as the checkpoints for accuracy assessment, …" in lines 434~436. Finally, the accuracy of the GCPs' 3D coordinates measured in the field, i.e., "the RTK-DGPS with a nominal positioning accuracy of 2 cm was used to measure the 3D coordinates of these GCPs" was reported in lines 429~430.

In addition, we replaced the word "surveys" with "investigation" in line 25.

We have tried our best to revise our manuscript according to your valuable comments, and hope that the correction will meet with approval.

[revised manuscript text omitted]

---

## Author Response (AR3)

*Response to EC:*

Authors: Thanks very much for your kind work. The respond to your comment is as flowing:

Dear Authors,
Thank you for your last version of the paper. Before publishing, I would ask you to improve Figure 4. At the moment it is still quite low resolution and many details are not completely visible.
After this improvement, the paper will be accepted.

Authors: Dear Editors, we have re-processed Figure 4, except that no other changes have been made, and hope that the correction will meet with approval.

[revised manuscript text omitted]

---

## Author Response (AR4)

*Response to EC:*

Authors: Thanks very much for your kind work. The respond to your comment is as flowing:

Dear Authors,
Editor Decision: Publish subject to technical corrections (20 Sep 2017) by F. Nex
Comments to the Author:
Dear authors,
the paper has been modified according to the requests of the referees. I would ask you to check for typos still present in the text. After this final check, the paper will be ready to be published.

Authors: Dear Editors, We have carefully checked all the typos and made the following modifications:

Line 4: Replace "The" with "the", and "province" with "Province";

Line 35: Replace semicolon (";") with comma (",") ;

Line 278: Replace "GPS" with "GNSS";

Lines 590-591, 598-601, 615-616, 619-623, 631-633, 637, 646, 656-657: Modify the reference spelling, to make more standardized.

We hope that the correction will meet with approval.

[revised manuscript text omitted]